# Balancing Interference and Correlation in Spatial Experimental Designs: A Causal Graph Cut Approach

Jin Zhu [* 1]  Jingyi Li [* 2]  Hongyi Zhou [3]  Yinan Lin [4]  Zhenhua Lin [2]  Chengchun Shi [1]

## Abstract

This paper focuses on the design of spatial experiments to optimize the amount of information derived from the experimental data and enhance the accuracy of the resulting causal effect estimator. We propose a surrogate function for the mean squared error (MSE) of the estimator, which facilitates the use of classical graph cut algorithms to learn the optimal design. Our proposal offers three key advances: (1) it accommodates moderate to large spatial interference effects; (2) it adapts to different spatial covariance functions; (3) it is computationally efficient. Theoretical results and numerical experiments based on synthetic environments and a dispatch simulator that models a city-scale ridesharing market, further validate the effectiveness of our design. A python implementation of our method is available at https://github.com/Mamba413/CausalGraphCut.

## 1. Introduction

**Background**. Before deploying any policy in practice, it is essential to evaluate its impact. This renders randomized control trials or online experiments extensively used for accurate policy evaluation. In numerous applications, the experiments involve multiple units that are either distributed across different spatial regions or connected through a network. The spatial or network dependencies inherent in these experiments pose two unique challenges for policy evaluation: (i) the existence of **interference effects** where interventions applied to one experimental unit may influence others, leading to the violation of the fundamental stable unit treatment value assumption (SUTVA) for causal inference (Imbens & Rubin, 2015); (ii) the strong **outcome correlations** among units, which increase the variance of the causal effect estimator. We present the following examples to illustrate.

**Example 1: A/B testing in marketplaces.** A/B testing is frequently used in two-sided markets such as Uber, Lyft, Airbnb and eBay to assess the impact of a newly developed product relative to a standard control. It has become the gold standard for companies in conducting data-driven decision making (Johari et al., 2022). Spatial interference is ubiquitous in marketplaces. For instance, ridesharing companies constantly offer subsidizing policies to drivers or passengers across different regions in a city. Applying a subsidizing policy in one location can attract drivers from neighboring regions to this area, potentially decreasing the driver supply in other regions. Thus, a subsidizing policy in one location can affect outcomes at other locations, inducing interference over space (Shi et al., 2023a). Meanwhile, market features like online driver numbers and call order numbers form interconnected spatial networks, leading to considerable spatial correlation (Luo et al., 2024).

**Example 2: Environmental and epidemiological applications.** Environmental and epidemiological studies frequently utilize spatial data to examine causal effects of certain interventions on health outcomes in different geographical regions (Reich et al., 2021). Spatial correlation and interference are likely to arise in these applications. For instance, in evaluating the impact of vaccination campaigns on disease incidence, infection rates exhibit strong correlations within regions. Furthermore, the vaccination coverage in a given region not only influences its own infection rate but also those in the neighboring regions (VanderWeele et al., 2012; Perez-Heydrich et al., 2014).

**Example 3: Experimentation in social networks.** Social network sites such as Facebook and LinkedIn extensively conduct online experiments to evaluate user engagement or satisfaction from a new service or feature. In these experiments, the treatment applied to a given user may spill over to other users via underlying social connections

---

[*]Equal contribution [1]Department of Statistics, London School of Economics and Political Science, London, United Kingdom [2]Department of Statistics and Data Science, National University of Singapore, Singapore [3]Department of Mathematics, Tsinghua University, Beijing, China [4]National Center for Applied Mathematics in Chongqing, Chongqing Normal University, Chongqing, China. Correspondence to: Zhenhua Lin <linz@nus.edu.sg>, Chengchun Shi <c.shi7@lse.ac.uk>.

*Proceedings of the 42nd International Conference on Machine Learning*, Vancouver, Canada. PMLR 267, 2025. Copyright 2025 by the author(s).

(like friends), leading to network interference. Additionally, since connected users often exhibit similar behaviors, their outcomes are likely correlated (Gui et al., 2015).

**Contributions**. This paper primarily focuses on the first application, aiming to identify the optimal design that minimizes the mean squared error (MSE) of the resulting average treatment effect (ATE) estimator. Nonetheless, the theories and methods developed herein can be adapted to the other two examples.

**Methodologically**, our contribution lies in the development of a causal graph cut algorithm for designing experiments with spatial interference and correlation. The advances of our algorithm are summarized as follows:

- **Flexibility**: Our proposal is more flexible than existing works that assume weak interference (Viviano et al., 2023). *It accommodates moderate to large interference effects*.

- **Adaptivity**: Instead of employing existing minimax approaches (Leung, 2022; Viviano et al., 2023; Zhao, 2024b) that minimize the same worst-case MSE across applications with different covariance structures and can be overly conservative, *our algorithm adapts to different correlation structures*, providing a tailored solution to each individual application.

- **Computational efficiency**: The design problem requires allocating treatment for each region and is inherently NP-hard, as the number of treatment allocation rules grows exponentially fast with respect to the number of regions. Unlike existing solutions that rely on computationally intensive integer programming (Zhao, 2024b), *we propose a new surrogate function for the MSE, which enables the use of spectral clustering-based graph cut algorithms for minimization*, thus substantially enhancing computational efficiency.

**Theoretically**, our analysis reveals that *the two pivotal factors — interference and correlation — drive the optimal design in opposite directions*. Specifically, experiments with large interference effects are best designed by assigning same policies to neighboring regions whereas those with strong spatial correlations benefit from allocating different policies. By incorporating the effects of the two factors into our surrogate function, the proposed algorithm effectively address both challenges, achieving desirable properties.

**Empirically**, based on synthetic environments and a city-level simulator — constructed using physical models and a real dataset from a ridesharing company to realistically simulate driver and passenger behaviors, we demonstrate the superior performance of our design compared to existing state-of-the-art methods. Notably, in the simulator, *the proposed ATE estimator achieves an MSE that is 3.5 times smaller than those of the benchmark approaches*.

## 1.1. Related works

This section reviews related works on spatial causal inference, experimental design and off-policy evaluation.

**Spatial causal inference**. Our proposal is closely related to a growing line of research on causal inference in the presence of spatial interference. Depending on the underlying assumptions, the methodologies developed within this area can be broadly grouped into three categories:

1. The first category assumes **partial interference** where units are partitioned into clusters and interference is restricted to within these clusters, not between them (Halloran & Struchiner, 1995; Sobel, 2006; Hudgens & Halloran, 2008; Tchetgen & VanderWeele, 2012; Crépon et al., 2013; Liu et al., 2016; Zigler & Papadogeorgou, 2021).

2. The second category assumes **neighborhood interference** where interference is limited to a unit's neighbors or a predefined local region (Verbitsky-Savitz & Raudenbush, 2012; Bhattacharya et al., 2020; Fatemi & Zheleva, 2020; Ma & Tresp, 2021; Gao & Ding, 2023; Dai et al., 2024; Jiang et al., 2024; Yang et al., 2024).

3. The last category considers more **general interference** structures (Aronow & Samii, 2017; Forastiere et al., 2021; Puelz et al., 2022; Song & Papadogeorgou, 2024; Zhang et al., 2024; Zhan et al., 2024).

In the machine learning literature, spatial causal inference is also related to recent advances on bandits with interference (Jia et al., 2024; Agarwal et al., 2024; Xu et al., 2024). However, despite the development of various methods to handle interference, experimental designs have been less considered in these papers.

**Experimental design under interference**. The design of experiments is a classical problem in statistics, motivated by applications ranging from biology and agriculture to psychology and engineering (Fisher et al., 1966). Recently, it has gained considerable attention in the field of machine learning (see e.g., Foster et al., 2021; Blau et al., 2022; Weltz et al., 2023; Fiez et al., 2024; Kato et al., 2024).

Our paper contributes to a growing line of research focusing on identifying optimal treatment assignment strategies in the presence of interference. Several designs have been proposed to guide treatment allocation in settings with interference over time (Hanna et al., 2017; Hu & Wager, 2022; Zhong et al., 2022; Bojinov et al., 2023; Li et al., 2023; Xiong et al., 2024; Sun et al., 2024; Wen et al., 2025) or among entities in marketplaces (Wager & Xu, 2019; Johari et al., 2020; Bajari et al., 2021; Munro et al., 2021; Li et al., 2022; Zhu et al., 2024).

This work studies designs under spatial or network interference. Methodologies developed within this field can be broadly categorized into three types:

1. The first type of methods adopts the principle of balance in the design of experiments. These methods focus on balancing covariates (Kallus, 2018; Liu et al., 2024; Harshaw et al., 2024) and cluster sizes in cluster-randomized designs (Gui et al., 2015; Saveski et al., 2017; Rolnick et al., 2019).

2. The second type of methods analytically calculates the MSE of the ATE estimator and uses it either to heuristically guide the design of cluster-randomized experiments (Ugander et al., 2013) or to determine the optimal number of clusters by minimizing the MSE's order of magnitude (Leung, 2022; Jia et al., 2023).

3. The last type of methods identifies the optimal design through an optimization perspective (Zhao, 2024a). They apply optimization tools to directly minimize the MSE of the treatment effect estimator or its proxy (Baird et al., 2018; Ni et al., 2023; Jiang & Wang, 2023; Viviano et al., 2023; Chen et al., 2024). In particular, Viviano et al. (2023) markedly advanced the design of cluster-randomized experiments under network interference by proposing a causal clustering algorithm, which offers a rigorous framework to numerically solve the optimal design via graph clustering algorithms. Subsequent developments are explored in several recent studies (see e.g., Eichhorn et al., 2024; Zhao, 2024b).

Our paper falls into the third category: it utilizes classical graph cut algorithms for optimization. Unlike Viviano et al. (2023) that employs a minimax formulation along with a weak interference assumption to simplify the MSE — potentially at the cost of being conservative and restrictive — our proposed algorithm does not rely on minimax formulations or limit itself to weak interference. Despite the more complicated form of the MSE, we identify a suitable surrogate to optimize, backed by theoretical guarantees. Empirically, our proposal substantially outperforms both causal clustering and other state-of-the-art in our application.

**Off-policy evaluation (OPE).** Finally, our work is also closely related to OPE in contextual bandits and reinforcement learning. OPE aims to learn the expected return under a target policy using an offline dataset collected from a possibly different behavior policy. There are three commonly-used OPE methods:

1. Direct method derives the policy value estimator by learning a value function that measures the (cumulative) reward under the target policy (Hahn, 1998; Le et al., 2019; Feng et al., 2020; Luckett et al., 2020; Hao et al.,

2021; Chen & Qi, 2022; Shi et al., 2022; Bian et al., 2024).

2. Importance sampling (IS) method reweights the observed reward using IS ratios to address the distributional shift between target and behavior policies (Heckman et al., 1998; Zhao et al., 2012; Swaminathan & Joachims, 2015; Thomas et al., 2015; Liu et al., 2018; Dai et al., 2020; Luckett et al., 2020; Kuzborskij et al., 2021; Wang et al., 2023; Hu & Wager, 2023; Zhou et al., 2025).

3. Doubly robust (DR) method and its variant combine the direct and IS estimators to achieve more accurate and robust estimation (Tan, 2010; Dudík et al., 2011; Zhang et al., 2012; 2013; Jiang & Li, 2016; Thomas & Brunskill, 2016; Chernozhukov et al., 2018; Liu et al., 2019; Chernozhukov et al., 2022; Shi et al., 2021; Kallus & Uehara, 2022; Liao et al., 2022; Xu et al., 2023; Shi et al., 2024).

Moreover, there is a growing line of research that proposes to adapt OPE methodologies to A/B testing (see e.g., Farias et al., 2022; Shi et al., 2023b). However, none of these aforementioned studies address interference or focus on experimental design. While works such as Wan et al. (2022), Hanna et al. (2017) and Liu & Zhang (2024) consider the design problem to optimize the efficiency of the OPE estimator, they do not address spatial/network interference either.

## 2. Preliminaries

In this section, we first formulate the A/B testing problem in spatial experiments, presenting our model and assumptions. We next detail the ATE estimator. Finally, we introduce cluster-randomized designs, which are extensively used in the field (see e.g., Ugander et al., 2013; Karrer et al., 2021; Leung, 2022).

**Spatial A/B testing**. We consider a spatial setting with $R$ non-overlapping regions across a city. The goal of spatial A/B testing is to compare the impact of implementing a newly developed policy in the whole city against an existing practice. Its procedure can be summarized as follows:

- At the beginning of the experiment, we observe a covariate matrix $\boldsymbol{O} = (O_1, \cdots, O_R)^\top \in \mathbb{R}^{R \times d}$ where each vector $O_i \in \mathbb{R}^d$ measures certain market features from the $i$th region. For instance, in ridesharing, $O_i$ could represent the number of initial drivers within that region.

- During the experiment, the decision maker assigns policies by specifying a vector $\boldsymbol{A} = (A_1, A_2, \ldots, A_R)^\top$ where each $A_i$ is a binary variable indicating whether the $i$-th region receives the newly developed policy ($A_i = 1$)

or a standard control policy ($A_i = 0$). In the context of ridesharing, $A_i$ could correspond to whether to assign certain driver-side or passenger-side subsidizing policy at the $i$th region (Shi et al., 2023a; Li et al., 2024).

- After the experiment, we collect an outcome vector $\boldsymbol{Y} = (Y_1, \cdots, Y_R)^\top$ with each $Y_i$ measuring the $i$-th region's outcome of interest (such as the total driver income). Throughout this paper, we assume $Y_i = g_i(\boldsymbol{A}, \boldsymbol{O}) + e_i$ for some unknown function $g_i$ and mean-zero error $e_i$ independent of $\boldsymbol{O}$ and $\boldsymbol{A}$.

- Finally, the experiment is repeated $N$ times, resulting in $N$ independent $(\boldsymbol{O}, \boldsymbol{A}, \boldsymbol{Y})$ triplets. Based on the data, we aim to estimate the ATE, defined as

$$\text{ATE} = \mathbb{E}[\text{CATE}(\boldsymbol{O})] = \sum_{i=1}^{R} \mathbb{E}[g_i(\boldsymbol{1}, \boldsymbol{O}) - g_i(\boldsymbol{0}, \boldsymbol{O})],$$

which measures the difference in outcome between applying the new policy globally to all regions and implementing the control policy. Here, CATE denotes the conditional ATE as a function of the observation matrix.

We make two remarks. First, unlike existing works on the design of spatial/network experiments (see Section 1.1) which analyze data from a single experiment without repeated measures, we consider settings with repeated measurements (Zhang & Wang, 2024). This is motivated by our ridesharing example where the experiment is conducted over two weeks, and each day's data can be treated as an independent realization since the number of call orders typically wanes between 1 and 5 am (Luo et al., 2024).

Second, our outcome regression model $Y_i = g_i(\boldsymbol{A}, \boldsymbol{O}) + e_i$ manifests the two challenges in spatial A/B testing:

1. **Interference**: $g_i$ depends not only on the $i$th region's own treatment $A_i$, but also on the treatments assigned to other regions as well.

2. **Correlation**: The covariance matrix of the residual $\boldsymbol{e} = (e_1, \cdots, e_R)^\top$, denoted by $\boldsymbol{\Sigma} \in \mathbb{R}^{R \times R}$, is typically non-diagonal, indicating that residuals are correlated across regions.

**Estimation**. To simplify the estimation under spatial interference, we impose the neighborhood interference assumption introduced in Section 1.1, which restricts the interference to neighboring regions. It is a specific yet widely used exposure mapping assumption (Aronow & Samii, 2017).

**Assumption 1** (Neighborhood interference). *For each region $i$, denote $\mathcal{N}_i$ as the set of spatial neighboring regions for the $i$-th region, including the $i$-th region itself. Let $\boldsymbol{a}_{\mathcal{N}_i}$ be the $|\mathcal{N}_i|$-dimensional subvector of $\boldsymbol{a}$ that is formed by the treatments assigned to the regions in $\mathcal{N}_i$. For any $i \in \{1, 2, \cdots, R\}$, $\boldsymbol{a}, \boldsymbol{a}' \in \{0, 1\}^R$ and $\boldsymbol{o} \in \mathbb{R}^d$, $g_i(\boldsymbol{a}, \boldsymbol{o}) = g_i(\boldsymbol{a}', \boldsymbol{o})$ whenever $\boldsymbol{a}_{\mathcal{N}_i} = \boldsymbol{a}'_{\mathcal{N}_i}$.*

For any $1 \le t \le n$ and $1 \le i \le R$, let $(A_{i,t}, Y_{i,t})$ denote the policy-outcome pair associated with the $i$-th region collected from the $t$-th experiment. Assumption 1 motivates us to consider the following importance sampling (IS) estimator popularly employed in the literature to estimate the ATE (see e.g., Ugander et al., 2013; Zhao, 2024b),

$$\widehat{\text{ATE}}^{\text{IS}} = \frac{1}{N} \sum_{t=1}^{N} \sum_{i=1}^{R} \left( \frac{T_{i,t}(\boldsymbol{1})}{\mathbb{E}[T_{i,t}(\boldsymbol{1})]} - \frac{T_{i,t}(\boldsymbol{0})}{\mathbb{E}[T_{i,t}(\boldsymbol{0})]} \right) Y_{i,t},$$

where $T_{i,t}(\boldsymbol{a}) := \prod_{j \in \mathcal{N}_i} \mathbb{I}(A_{j,t} = a_j)$ so that $T_{i,t}(\boldsymbol{1})$ (or $T_{i,t}(\boldsymbol{0})$) indicates whether all neighbors of the $i$-th region (including itself) are treated with the new policy (or the standard control).

A well-known limitation of IS estimators is its large variance (Dudík et al., 2011). To mitigate this issue, we employ the following doubly robust (DR) estimator developed by Yang et al. (2024) for ATE estimation. DR attains theoretically no larger and empirically often smaller MSE than IS, and is defined as

$$\widehat{\text{ATE}}^{\text{DR}} = \frac{1}{N} \sum_{t=1}^{N} \sum_{i=1}^{R} [\nu_{i,t}(\boldsymbol{1}) - \nu_{i,t}(\boldsymbol{0})], \tag{1}$$

where $\nu_{i,t}(\boldsymbol{a}) = g_i(\boldsymbol{a}, \boldsymbol{O}_t) + \frac{T_{i,t}(\boldsymbol{a})}{\mathbb{E}[T_{i,t}(\boldsymbol{a})]}[Y_{i,t} - g_i(\boldsymbol{a}, \boldsymbol{O}_t)]$ and $\boldsymbol{O}_t$ denotes the observation matrix from the $t$-th experiment. Notice that $\nu_{i,t}$ depends on $g_i$, which is unknown and must be estimated. In practice, estimation of $g_i$ can be achieved either parametrically, using a specified functional form based on lower-dimensional statistics of actions and observations (Hu et al., 2022), or nonparametrically through neural networks (Leung & Loupos, 2022; Dai et al., 2024; Wang et al., 2024). Thanks to its double robustness property and the fact that the randomization probability $\mathbb{E}[T_{i,t}(\boldsymbol{a})]$ is known by design, the resulting estimator remains unbiased even when the estimated $g_i$ is inconsistent. Notably, when the estimated $g_i$ is set to zero, the resulting estimator simplifies to IS.

**Design**. In our context, each design specifies a treatment allocation rule that determines the joint probability distribution function of $\boldsymbol{A}$. Following the existing practice, we focus on the class of *cluster-randomized designs* in this paper. These designs partition the $R$ regions into $m$ disjoint clusters $\mathcal{C}_1, \ldots, \mathcal{C}_m$ where regions within the same cluster receive the same treatment. Meanwhile, treatments across different clusters are independent, each following a Bernoulli distribution with a probability $p$ of receiving the new policy. We further fix $p = 0.5$ throughout the paper to ensure a *balanced design*, since the optimal design is indeed balanced (Yang et al., 2024).

Two special cases are worthwhile to mention: (i) When $m = 1$, there is only one cluster. The resulting design is

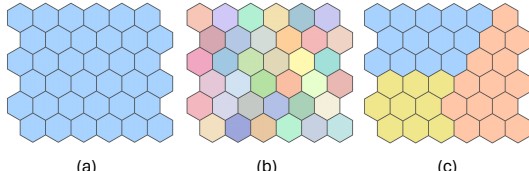

(a)  (b)  (c)

*Figure 1.* Illustrations of the optimal cluster-randomized designs in three experiments, where different colors represent different clusters. (a) With weakly correlated residuals, the optimal design simplifies to the global design. (b) Without interference, the optimal design reduces to the individual design. (c) With moderate to large levels of interference and correlation, the optimal design falls between the two extremes, resulting in a 3-cluster design.

reduced to a *global design* where all regions receive the same policy during each experiment. (ii) When $m = R$, each region forms a single cluster, resulting in an *individual design* where policies are *i.i.d.* across regions.

Our objective lies in determining the optimal cluster-randomized design, for which we propose a causal graph cut algorithm to minimize the MSE of the ATE estimator in (1). Details are provided in the next section.

## 3. Causal Graph Cut

This section is organized as follows. Section 3.1 analyzes the roles of spatial interference and correlation in designing cluster-randomized experiments. Section 3.2 derives our surrogate function for the MSE of the ATE estimator. Finally, Section 3.3 details our causal graph cut algorithm.

### 3.1. Cluster-randomized design: the roles of interference and correlation

We will show that interference and correlation are the two driving factors in determining the MSE of ATE estimators under different designs. Interestingly, they influence the design of experiments in completely different directions:

**Claim 1.** *Experiments with **large interference** are best designed by applying **same policies** to neighboring regions.*

**Claim 2.** *Experiments with **strong correlation** are best designed by allocating **different policies** to different regions.*

To illustrate these two claims, we consider three experiments visualized in Figure 1, with the same spatial grid. They differ in the sizes of interference effects and correlation: (i) In the left experiment, there is moderate to large interference, while the residuals $e_1, \ldots, e_R$ are *weakly* correlated across regions. (ii) In the middle experiment, SUTVA holds, meaning there is *no interference*, but there is moderate to large correlation. (iii) In the right experiment, both levels of interference and correlation range from *moderate to large*.

Figure 1 further visualizes the optimal cluster-randomized

designs for the three experiments, with different colors representing different clusters. It clearly shows that:

(i) With weakly correlated errors, interference plays the leading role in determining the MSE. Since all regions are connected, according to Claim 1, the optimal allocation rule assigns the same policy to all regions, resulting in a global design.

(ii) Without interference effects, correlation becomes the sole factor in determining the MSE. According to Claim 2, the optimal allocation rule assigns different treatments across regions, creating an individual design.

(iii) When both interference and correlation are present, the optimal design typically falls between the two extremes, resulting in a cluster-randomized design. Specifically, in the right experiment, the proposed algorithm identifies a three-cluster design as the optimal design.

For any two regions $i$ and $i'$, let $\Sigma_{ii'}$ denote the covariance between two residuals $e_i$ and $e_{i'}$. Let $\delta > 0$ represent the size of the spatial correlation so that $|\Sigma_{ii'}| \leq \delta$ for any two non-neighboring regions $i$, $i'$. The following two propositions formally summarize the aforementioned findings.

**Proposition 1.** *When $\delta$ is sufficiently small (i.e., spatial correlation is weak among non-neighboring regions), and the covariance $\Sigma_{ii'}$ between any two neighboring regions $i, i'$ is positive, the global design minimizes the MSE of the DR estimator* (1).

**Proposition 2.** *When SUTVA (i.e., the no-interference assumption) holds and all entries of $\mathbf{\Sigma}$ are non-negative, the individual design minimizes the MSE of the DR estimator.*

To verify the two claims and propositions, we offer a decomposition of the DR estimator's MSE in the following theorem. Recall that $N$ denotes the number of repeated experiments. For a given cluster $\mathcal{C}$, let $\partial\mathcal{C}$ denote its boundary.

**Theorem 1.** *Under Assumption 1, the DR estimator's MSE under a cluster-randomized design with $m$ clusters $\{\mathcal{C}_j\}_{j=1}^m$ is given by*

$$\mathrm{MSE}(\widehat{\mathrm{ATE}}^{\mathrm{DR}}) = \underbrace{\frac{1}{N}\mathrm{Var}\left\{\mathrm{CATE}(\boldsymbol{O}_t)\right\}}_{\text{design-agnostic (DA) term}} + \underbrace{\frac{4}{N}\left[\sum_{j=1}^m \sum_{i,i' \in \mathcal{C}_j} \Sigma_{ii'}\right]}_{\text{spatial correlation (SC) term}}$$

$$+ \underbrace{\frac{8}{N}\left[\sum_{j \neq k}\sum_{i \in \mathcal{C}_j}\sum_{i' \in \partial\mathcal{C}_k} \Sigma_{ii'}\mathbb{I}(\mathcal{N}_{i'} \cap \mathcal{C}_j \neq \emptyset)\right]}_{\text{first-order interference term } I_1}$$

$$+ \underbrace{O\left[\frac{1}{N}\sum_{j,k=1}^m \sum_{i \in \partial\mathcal{C}_j}\sum_{i' \in \partial\mathcal{C}_k} |\Sigma_{ii'}|\right]}_{\text{second-order interference term } I_2},$$

*where the event $\mathcal{N}_{i'} \cap \mathcal{C}_j \neq \emptyset$ indicates that the $i'$-th region is adjacent to the $j$-th cluster.*

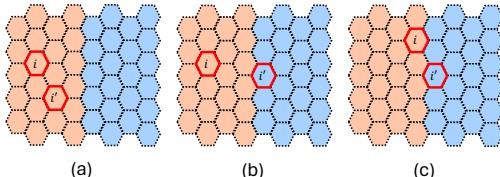

Figure 2. An illustration of each term in the MSE decomposition from Theorem 1 with $m = 2$ clusters distinguished by different colors. (a) SC captures the residual covariance between any $i, i'$ within the same cluster. (b) $I_1$ captures the residual covariance between any $i, i'$ that belong to different clusters, with one region lying on the boundary between the two clusters. (c) $I_2$ captures residual covariance between any $i, i'$ from different clusters, with both regions located on the boundary.

Theorem 1 decomposes the MSE into four terms. We elaborate each of them below.

1. The **design-agnostic term** (DA) accounts for the variation in covariates across $N$ experiments. It is independent of the design and hence *design-agnostic*.

2. The **spatial correlation term** (SC) captures the residual covariances between regions within the same cluster, as illustrated in Figure 2(a). When the covariance function is non-negative, it becomes evident that minimizing this term is equivalent to assigning different policies to different regions, thus demonstrating Claim 2. Notably, it attains the minimum value under the individual design, which in turn proves Proposition 2.

3. The **interference terms** $I_1$ and $I_2$ equal zero in the absence of spatial interference. Unlike the SC which captures only within-cluster correlation, both $I_1$ and $I_2$ account for between-cluster correlation, as illustrated in Figure 2(b) and (c). Under neighborhood interference, the first-order term $I_1$ requires at least one region to lie on the boundaries between the two clusters, representing a *first-order* boundary effect. The second-order term $I_2$ further requires that both regions be located on the boundary, thus characterizing a *second-order* boundary effect. For any two neighboring regions, assigning them different policies causes them to belong to connected yet different clusters, which creates boundaries. Thus, minimizing these terms is equivalent to assigning the same policy to neighboring regions, which demonstrates Claim 1. Apparently, there is no boundary effect under the global design. This proves Proposition 1.

In summary, interference and correlation drive the optimal design in opposite directions. Interference induces estimation errors at cluster boundaries. Thus, assigning the same policy to neighboring regions eliminates the boundary effect and minimizes these errors. On the contrary, allocating different policies to different regions effectively negates the positive correlation between their residuals, thus reducing the estimation errors caused by spatial correlation.

### 3.2. Balancing interference and correlation: a new surrogate function for the MSE

In this section, we design a surrogate function aimed at achieving the following properties:

**Property 1** (Flexibility). *The surrogate function shall accommodate moderate to large interference effects.*

**Property 2** (Adaptivity). *The surrogate function shall adapt to different correlation structures.*

**Property 3** (Computational efficiency). *The surrogate function can be optimized efficiently.*

Flexibility is crucial, since limiting the methodology to weak interference would be overly restrictive. Additionally, computational efficiency is essential for the practical implementation of the design. Below, we provide a numerical example to highlight the importance of adaptivity.

**A numerical example**. Consider a square spatial grid with $R = 144$ regions visualized in Figure 3(a) and an exponential spatial correlation structure: $\Sigma_{ij} = \rho^{|i-j|}$ for some $0 < \rho \le 1$ that characterizes the strength of spatial correlation. We numerically compute MSEs of DR estimators under different cluster-randomized designs, each dividing the spatial grid into $m$ equally sized square rectangles (see Figure 3(a) for an example with $m = 4$). Notice that varying $\rho$ induces different covariance functions whereas varying $m$ yields different designs. We report the MSE for each combination of $\rho$ and $m$ in Figure 3(b), where the MSE values on the $y$-axis are presented using a logarithmic scale for clearer visualization and comparison. These results emphasize two key motivations, which we discuss below.

1. We first note that the MSE varies considerably with cluster size. Notably, when $\rho = 0.9$, the MSE reaches its minimum value of 20 with 4 clusters, but increases to over 40 under the global design. *This highlights the motivation for addressing the design problem itself*.

2. We also observe that the optimal cluster size varies with $\rho$. *This highlights the motivation for developing an adaptive method capable of identifying the optimal design tailored to each individual covariance function.*

**A new surrogate function**. It remains challenging to design a surrogate function for the MSE that satisfies the aforementioned three properties. Under neighborhood interference, the primary challenge arises from the presence of the two interference terms $I_1$ and $I_2$ (see Theorem 1), the calculation of which requires specifying cluster boundaries. Since clusters under different designs have different boundaries, optimizing these terms becomes an NP-hard combinatorial optimization problem, conflicting with Property 3. One solution is to impose the weak interference assumption as in Viviano et al. (2023), under which

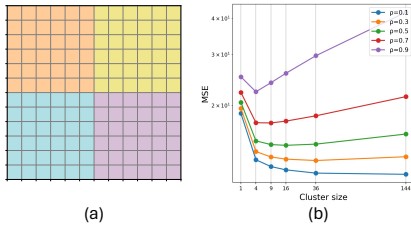

(a)

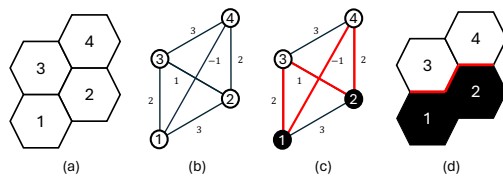

*Figure 3.* (a) An $12 \times 12$ grid divided into four square clusters. (b) A scatter plot visualizing the DR estimator's MSE as a function of cluster size. $\rho$ is selected from $\{0.1, 0.3, 0.5, 0.7, 0.9\}$.

the boundary effect becomes negligible (see their proof of Lemma 3.2). However, this approach violates Property 1.

For illustration purposes, we will now focus on designs with two clusters to discuss how we address these challenges and detail our surrogate function. Its more general form, applicable to general numbers of clusters, is presented in Appendix A. Let $\mathcal{C}_1$ and $\mathcal{C}_2$ denote a partition of all $R$ regions, our surrogate function is defined as follows:

$$\frac{8R}{N} \sum_{i \in \mathcal{C}_1} \sum_{i' \in \mathcal{C}_2} W_{ii'} \Sigma_{ii'}^+ - \frac{8}{N} \sum_{i \in \mathcal{C}_1} \sum_{i' \in \mathcal{C}_2} \Sigma_{ii'}, \quad (2)$$

where $\{W_{ii'}\}_{i,i'=1,...,R}$ denotes the binary adjacency matrix indicating whether two regions are adjacent, and $\Sigma_{ii'}^+ = \max(\Sigma_{ii'}, 0)$. This objective function is built upon Theorem 1 and is motivated by the following observations:

1. The DA term does not need to be incorporated in the surrogate, since it is design-agnostic.

2. Minimizing the SC term, i.e., the within-cluster covariance, is equivalent to maximizing the between-cluster covariance, represented by the second term in (2).

3. Instead of directly minimizing $I_1$, we use a surrogate upper bound given by the first term in (2) that does not involve cluster boundaries and can be more efficiently optimized.

4. Applying a similar upper bound to $I_2$ will result in a quartic objective function, rather than quadratic, rendering spectral clustering-based cut algorithms inapplicable. However, as indicated by Figure 2(c), only a small fraction of the regions lie at the boundary, making $I_2$ a high-order and negligible term when compared with $I_1$. This justifies omitting $I_2$ from the objective function.

Proposition 3 below formally establishes that the first term in (2) serves as a valid upper bound for $I_1$, under the following assumption which is automatically satisfied when the covariance function decays with distance — a condition commonly imposed in spatial statistics (Cressie, 2015).

**Assumption 2** (Decaying covariance). *For any three disjoint regions $i_1$, $i_2$, $i_3$ such that only $i_1$ and $i_2$ are neighbors, it holds that $\Sigma_{i_1,i_2} \geq \Sigma_{i_1,i_3}$.*

*Figure 4.* (a) Spatial layout of 4 regions to be partitioned. (b) Each region is represented as a vertex, with edges determined by the weight $\omega$. (c) The cut highlighted in red minimizes the total loss weight. (d) The spatial layout obtained after applying the 2-cut, illustrating how the regions are partitioned based on the graph cut.

**Proposition 3.** *Under Assumption 2, the first term in (2) upper bounds $I_1$.*

Meanwhile, when the covariance function is bounded away from zero, i.e., $\min_{ii'} \Sigma_{ii'} > 0$, the following Proposition shows that this upper bound remains tight in the sense that it is of the same order of magnitude as $I_1$.

**Proposition 4.** *When $\epsilon > 0$, the first term in (2) is upper bounded by $d_{\max} \sigma I_1$ where $d_{\max}$ denotes the maximum number of neighbors across regions and $\sigma$ denotes the ratio $\max_{i \neq i'} \Sigma_{ii'} / \min_{i,i'} \Sigma_{ii'}$.*

It is also worthwhile to mention that our surrogate function explicitly relies on the covariance function $\Sigma$, allowing the resulting design to be dependent upon $\Sigma$. Alternatively, one could employ a minimax approach similar to Viviano et al. (2023) and Zhao (2024b) that derives a worst-case surrogate among all possible covariance functions. However, the resulting design is no longer adaptive to $\Sigma$.

In summary, the two terms in (2) measure the effects of interference and correlation, respectively. Optimizing these terms drives the resulting designs as described in Claims 1 and 2. By incorporating both, our surrogate function balances interference and correlation, successfully achieving the first two desired properties. We will discuss how to efficiently optimize this function in the next section.

### 3.3. Causal graph cut: the detailed algorithm

We focus on the case with two clusters to illustrate our algorithm. Notice that the surrogate function in (2) is proportional to $\sum_{i \in \mathcal{C}_1} \sum_{i' \in \mathcal{C}_2} \omega_{ii'}$ where $\omega_{ii'} = RW_{ii'} \Sigma_{ii'}^+ - \Sigma_{ii'}$ can be viewed as the weight between any two regions. Assigning two regions to different clusters results in a loss of their weight, and minimizing the surrogate function is equivalent to determining the optimal clustering that minimizes the total lost weight. This formulation allows us to employ classical graph cut algorithms (see e.g., Goldschmidt & Hochbaum, 1994; Stoer & Wagner, 1997) to optimize our surrogate function, as illustrated in Figure 4.

Following Hagen & Kahng (1992), let $L$ represent the Laplacian matrix of $\Omega = \{\omega_{ii'}\}_{i,i'}$ and define the partition vector $x$ such that $x_i = 1/\sqrt{R}$ if $i \in \mathcal{C}_1$ and $x_i = -1/\sqrt{R}$ if $i \in \mathcal{C}_2$. A key observation is that the cut loss func-

tion can be represented using the following quadratic form $\boldsymbol{x}^\top L \boldsymbol{x}/4$. Instead of employing integer programming to optimize this quadratic form, we relax the search space to the entire unit ball to facilitate computation. This simplifies the optimization to identifying the eigenvectors of $L$. Given that the Laplacian matrix has a trivial eigenvector (a vector of all ones) with an associated eigenvalue of zero, classical graph cut algorithms typically compute the eigenvector associated with the second smallest eigenvalue, known as the Fiedler vector (Fiedler, 1973; 1989).

However, our problem sightly differs from the classical graph cut problem in that the weight matrix is not guaranteed to be positive definite. Consequently, if the smallest eigenvalue equals zero, we search for the Fiedler vector. Otherwise, we employ the first eigenvector. Once we have obtained the appropriate eigenvector, we apply the $k$-means algorithm to finalize the partition. In more general settings with $m > 2$ clusters, instead of searching for the first or second eigenvector, we search for a few (i.e., $\log_2(m)$ many), and apply the $k$-means algorithm to determine the $m$ clusters. By varying $m$ from 1 to a predefined maximum value $m_{\max}$ — set to $R^{2/3}$ in our implementation based on Leung (2022)'s recommendation — we obtain $m_{\max}$ many designs. We next evaluate their performance by plugging each design into the MSE formula derived from Theorem 1 (instead of the surrogate function) and choose the number of clusters that minimizes the MSE. Finally, the aforementioned discussion implicitly assumes that $\boldsymbol{\Sigma}$ is known; however, in practical applications, direct access to $\boldsymbol{\Sigma}$ is often not feasible. Since the experiment is repeated $N$ times, we can estimate $\boldsymbol{\Sigma}$ to construct the MSE estimator and update the design in an iterative manner. We summarize the procedure in Algorithm 1.

## 4. Experiments

In this section, we conduct numerical experiments to compare the ATE estimator computed via the proposed causal graph cut (CGC) algorithm against those constructed based on the following benchmark methods: (i) spectral clustering (SC, Leung, 2022); (ii) the three-net algorithm (TNET, Uganda et al., 2013); (iii) causal clustering (CC, Viviano et al., 2023). we also compare against the DR estimators constructed based on (iv) the global design (GD) and (v) the individual design (ID). We also compare against (vi) an oracle version of the proposed CGC algorithm (denoted by OCGC) which works as well as if the covariance matrix $\boldsymbol{\Sigma}$ is known in advance.

**Synthetic environments**. We design four spatial settings visualized in Figure 5(a), covering four classical urban morphology (Kropf, 2017). We set function $g_i$ to be a sinusoidal function that incorporates both spatial covariates and treatments, with correlation parameter $\rho$ indicating de-

---

**Algorithm 1** Causal graph cut (CGC)

**Input:** A batch sample size $B$ and an initial clustering $\mathcal{C}$.
1: **for** $l = 1, \ldots, N/B$ **do**
2:     Implement the design based on $\mathcal{C}$ and collect the dataset $\mathcal{D}^{(l)} = \{(\boldsymbol{Y}_t, \boldsymbol{A}_t, \boldsymbol{O}_t)\}_{t=1}^B$.
3:     Use datasets $\mathcal{D}^{(1)}, \ldots, \mathcal{D}^{(l)}$ to estimate $g_i$ (denoted by $\widehat{g}_i$) and compute $\widehat{e}_{it} = Y_{it} - \widehat{g}_i(\boldsymbol{A}_{\mathcal{N}_i,t}, \boldsymbol{O}_{\mathcal{N}_i,t})$.
4:     Compute the estimated covariance matrix $\widehat{\boldsymbol{\Sigma}}$ whose $(i, i')$-th entry is given by $\widehat{\Sigma}_{ii'} = \sum_{t=1}^B \widehat{e}_{it}\widehat{e}_{i't}/B$.
5:     **for** $m = 1, \cdots, m_{\max}$ **do**
6:         Use $\widehat{\boldsymbol{\Sigma}}$ to estimate the MSE and apply graph cut to learn the optimal design with $m$ clusters.
7:     **end for**
8:     Select the number of clusters whose clustering minimizes the estimated MSE and update $\mathcal{C}$.
9:     Calculate the DR estimator $\widehat{\text{ATE}}_l$ according to (1) using the data $\mathcal{D}^{(l)}$.
10: **end for**
**Output:** $\widehat{\text{ATE}} = (B/N) \sum_{l=1}^{N/B} \widehat{\text{ATE}}_l$

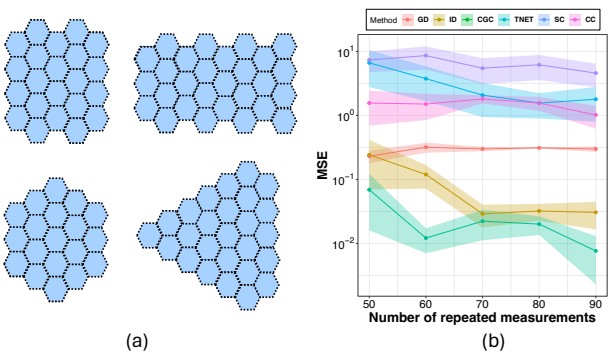

(a)                  (b)

*Figure 5.* (a) Four types of spatial grids in synthetic environments: square (top-left), rectangle (top-right), circle (bottom-left), fans (bottom-right). (b) MSEs with different numbers of repetitions $N$ in the real-data-based simulator. The shaded area visualizes the confidence interval.

---

gree of spatial dependence, evaluated under three different covariance structures (constant, truncated constant, exponential), see Appendix C.1 for further details. We study the MSE of estimators as $\rho$ increases from 0.1 to 0.9. The numerical results are shown in Figure 6, with values on the $y$-axis presented on a logarithmic scale for clearer visualization. It can be seen that our algorithms CGC and OCGC achieve MSEs that are substantially smaller than CC, TNET and SC. Besides, regardless of the spatial setting and covariance structure, our design consistently surpasses both the naive global design and individual design, with 54% and 72% improvements, which reflects the advantage of our proposed algorithms. Finally, it is worth mentioning that the CGC closely matches the performance of OCGC, indicating the effectiveness of our iterative framework.

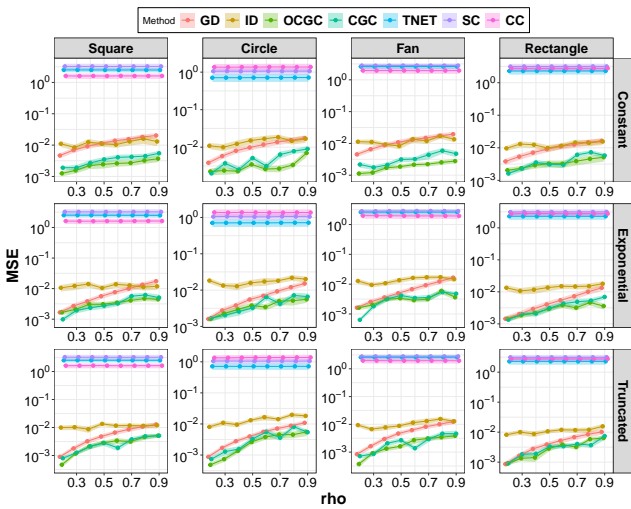
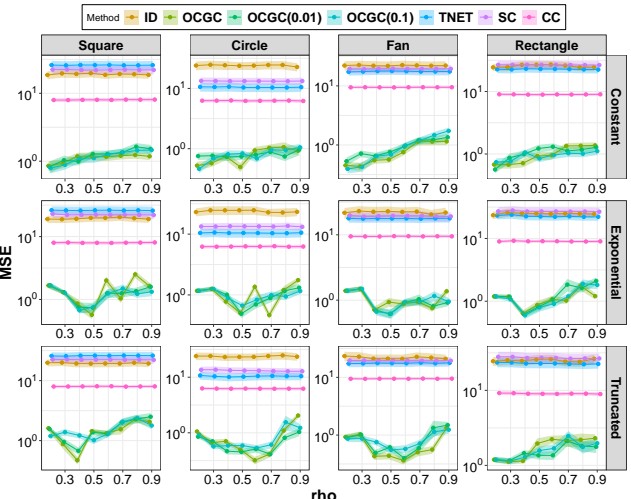

*Figure 6.* The plot of $\rho$ versus MSE, with each panel representing a different spatial arrangement setting.

*Figure 7.* The plot of $\rho$ versus MSE in single-experiment settings, with each panel representing a different spatial arrangement. OCGC (0.01) and OCGC (0.1) represent our method using a noised covariance matrix, where the noise follows a normal variable with mean 0 and variance 0.01 and 0.1, respectively. We did not implement GD in single-experiment settings, since it assigns the same treatment to all regions, making the ATE unidentifiable from the experimental data.

**Real-data-based simulator**. We develop a simulator using a five-day historical dataset from a ridesharing company to evaluate the proposed approach. This simulator replicates key dynamics of a city-scale ridesharing market with $R = 85$ hexagonal regions. Every 2 seconds, the simulator updates, among others: (1) drivers' decisions to accept assigned orders, taking into account historical driver and order characteristics; (2) driver movements to pick-up locations or idle movement patterns based on historical trajectories; (3) the driver pool, accounting for new active drivers and those going offline; and (4) new orders based on historical data, incorporating factors like passenger subsidies and processing unassigned and new requests. Particularly, drivers' decisions and reposition leverage the prediction of machine learning models trained with datasets that includes about 1 million observations. Furthermore, to match with the real world setting, when each simulation starts, drivers are distributed across the city based on their empirical distribution from the offline dataset that has more than 70,000 observations. See Appendix C.2 for more details.

In the experiments, the covariate is the number of initial drivers available at the start of each time period. The outcome, measured by gross merchandise value (GMV), will be used to evaluate the effectiveness of the policy. We exclude OCGC since we cannot assess the spatial correlation of the real-world environment. The results are presented in Figure 5(b).

First, we observe that our proposed method achieves the lowest MSE in most cases, with its MSE consistently decreasing as the number of repeated experiments increases, demonstrating the learning capability of our iterative framework. The improvement over the benchmarks is substantial, with the MSE being approximately 3.5 times

lower. Second, due to the strong spatial correlation in real-world environments, the global design is significantly less effective than the individual design, further supporting Proposition 1. The individual design, in turn, is substantially less effective than the proposed method, highlighting the advantages of our approach in balancing spatial correlation and interference.

**Single-experiment settings.** Finally, we remark that our core methodology does not rely on the assumption of repeated experiments. It remains applicable to single-experiment settings when we have certain prior knowledge regarding the underlying covariance matrix, either from a pilot study or based on historical data. To support this claim, we used a proxy covariance matrix constructed by adding noise to the true covariance matrix, and present the corresponding numerical results in Figure 7.

The results demonstrate that our estimator: (i) maintains optimality against existing methods; (ii) achieves near-oracle performance (comparable to the oracle method with the true covariance matrix); (iii) remains robust to the approximation errors in the covariance matrix.

We also remark that while cross-fitting helps simplify the theory (by avoiding the need for imposing VC-class conditions on to establish the asymptotics of the ATE estimator), our method remains effective without it - numerical results above were obtained without cross-fitting.

## Acknowledgements

Zhu's and Shi's research was supported by the EPSRC grant EP/W014971/1. Zhou's research was partially supported by the China Scholarship Council. Lin's research was partially supported by MOE AcRF Tier 1 grant. The authors express their gratitude for the constructive comments provided by the referees, which substantially enhanced the initial version of the paper.

## Impact statement

The broader impact of this work lies in its potential to enhance policy evaluation and decision-making across diverse domains, including urban planning, transportation, healthcare, and social sciences. Its scalability and adaptability provide opportunities to address complex societal challenges where spatial data plays a pivotal role. However, the application of our adaptive experimental design algorithm requires careful consideration of data privacy, transparency, and informed consent—especially in sensitive contexts involving personal or sensitive information—to ensure its responsible and ethical deployment.

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

## A. Surrogate function for general $m$

For the general case where $m > 2$, similarly, to minimize MSE that is affected by the clustering configuration, it is sufficient to minimize the upper bound of $SC + I_1$ by solving the following objective function:

$$\sum_{j<k}\sum_{i\in\mathcal{C}_j}\sum_{i'\in\mathcal{C}_k}\frac{2R}{m}\Sigma_{ii'}^+ W_{ii'} - \sum_{j<k}\sum_{i\in\mathcal{C}_j}\sum_{i'\in\mathcal{C}_k}\Sigma_{ii'}, \tag{3}$$

where the first term is an upper bound of $I_1$ and the later one is equal to the term SC after adding a constant $\sum_{i,i'}\Sigma_{ii'}$. The objective function (3) forms a graph-cut problem.

## B. Proofs

### B.1. Proof of Theorem 1

*Proof.* According to Theorem 2 in Yang et al. (2024) and assuming functions $g_i$ are known, we have

$$\mathrm{MSE}(\widehat{\mathrm{ATE}}^{\mathrm{DR}}) = \sigma_1^2 + \mathrm{DA},$$

where

$$\sigma_1^2 = \frac{1}{N}\sum_{i,i'}\left(\frac{1}{p^{m_{ii'}}} + \frac{1}{(1-p)^{m_{ii'}}}\right)\Sigma_{ii'}\mathbb{I}(m_{ii'} > 0),$$

$$\mathrm{DA} = \frac{1}{N}\mathrm{Var}\left\{\sum_{i=1}^{R}[g_i(\mathbf{1}, \mathbf{O}_t) - g_i(\mathbf{0}, \mathbf{O}_t)]\right\},$$

where $m_{ii'} = \sum_{j=1}^{m}\mathbb{I}(\mathcal{N}_i \cap \mathcal{C}_j \neq \varnothing)\mathbb{I}(\mathcal{N}_{i'} \cap \mathcal{C}_j \neq \varnothing)$ represents the number of clusters that both $\mathcal{N}_i$ and $\mathcal{N}_{i'}$ belong to.

We now elaborate that $\sigma_1^2 = SC + I_1 + I_2$. Note that we set $p = 0.5$ throughout this paper for balanced design. Let $\mathcal{C}_j^0$ denote the interior region of cluster $\mathcal{C}_j$. We first rewrite $\sigma_1^2$ in the following form:

$$\sigma_1^2 = \frac{1}{N}\mathrm{Var}\left[\sum_{j=1}^{m}\left(\sum_{i\in\mathcal{C}_j^0}\frac{2A_i-1}{p}e_i + \sum_{i\in\partial\mathcal{C}_j}\frac{\mathbb{I}(A_{\mathcal{N}_i}=1) - \mathbb{I}(A_{\mathcal{N}_i}=0)}{p^{m_i}}e_i\right)\right],$$

which can be decomposed into the sum of a within-cluster term $\sigma_w^2$ and a between-cluster term $\sigma_b^2$. Specifically,

$$\begin{aligned}
\sigma_w^2 &= \frac{1}{N}\left[\sum_{j=1}^{m}\sum_{i,i'\in\mathcal{C}_j^0}\frac{\Sigma_{ii'}}{p^2} + 2\sum_{j=1}^{m}\sum_{i\in\mathcal{C}_j^0}\sum_{i'\in\partial\mathcal{C}_j}\frac{\Sigma_{ii'}}{p^2} + \sum_{j=1}^{m}\sum_{i,i'\in\partial\mathcal{C}_j}\frac{\Sigma_{ii'}}{p^{1+m_{ii'}}}\right] \\
&= \frac{1}{N}\left[\sum_{j=1}^{m}\sum_{i,i'\in\mathcal{C}_j}\frac{\Sigma_{ii'}}{p^2} + \sum_{j=1}^{m}\sum_{i,i'\in\partial\mathcal{C}_j}\frac{\Sigma_{ii'}(1-p^{m_{ii'}-1})}{p^{1+m_{ii'}}}\right] \\
&= SC + \underbrace{\frac{1}{N}\left[\sum_{j=1}^{m}\sum_{i,i'\in\partial\mathcal{C}_j}\frac{\Sigma_{ii'}(1-p^{m_{ii'}-1})}{p^{1+m_{ii'}}}\right]}_{J_1},
\end{aligned}$$

and

$$\sigma_b^2 = \frac{2}{N}\Big[\sum_{j\neq k}\sum_{i\in\mathcal{C}_j^0}\sum_{i'\in\partial\mathcal{C}_k}\frac{\Sigma_{ii'}}{p^2}\mathbb{I}(\mathcal{N}_{i'}\cap\mathcal{C}_j\neq\emptyset)\Big] + \frac{1}{N}\Big[\sum_{j\neq k}\sum_{i\in\partial\mathcal{C}_j}\sum_{i'\in\partial\mathcal{C}_k}\frac{\Sigma_{ii'}}{p^{m_{ii'}+1}}\mathbb{I}(m_{ii'}>0)\Big]$$

$$= \frac{2}{N}\Big[\sum_{j\neq k}\sum_{i\in\mathcal{C}_j}\sum_{i'\in\partial\mathcal{C}_k}\frac{\Sigma_{ii'}}{p^2}\mathbb{I}(\mathcal{N}_{i'}\cap\mathcal{C}_j\neq\emptyset)\Big] + \frac{1}{N}\Big[\sum_{j\neq k}\sum_{i\in\partial\mathcal{C}_j}\sum_{i'\in\partial\mathcal{C}_k}\Big(\frac{\Sigma_{ii'}}{p^{m_{ii'}+1}} - \frac{2\Sigma_{ii'}}{p^2}\Big)\mathbb{I}(\mathcal{N}_{i'}\cap\mathcal{C}_j\neq\emptyset)\Big]$$

$$+ \frac{1}{N}\Big[\sum_{j\neq k}\sum_{i\in\partial\mathcal{C}_j}\sum_{i'\in\partial\mathcal{C}_k}\frac{\Sigma_{ii'}}{p^{m_{ii'}+1}}\Big(\mathbb{I}(m_{ii'}>0) - \mathbb{I}(\mathcal{N}_{i'}\cap\mathcal{C}_j\neq\emptyset)\Big)\Big]$$

$$= I_1 + \underbrace{\frac{1}{N}\Big[\sum_{j\neq k}\sum_{i\in\partial\mathcal{C}_j}\sum_{i'\in\partial\mathcal{C}_k}\Big(\frac{\Sigma_{ii'}}{p^{m_{ii'}+1}} - \frac{2\Sigma_{ii'}}{p^2}\Big)\mathbb{I}(\mathcal{N}_{i'}\cap\mathcal{C}_j\neq\emptyset)\Big]}_{J_2}$$

$$+ \underbrace{\frac{1}{N}\Big[\sum_{j\neq k}\sum_{i\in\partial\mathcal{C}_j}\sum_{i'\in\partial\mathcal{C}_k}\frac{\Sigma_{ii'}}{p^{m_{ii'}+1}}\Big(\mathbb{I}(m_{ii'}>0) - \mathbb{I}(\mathcal{N}_{i'}\cap\mathcal{C}_j\neq\emptyset)\Big)\Big]}_{J_3}.$$

We denote $I_2 = J_1 + J_2 + J_3$. Then we have $\sigma_1^2 = SC + I_1 + I_2$. $\hfill\square$

## B.2. Proof of Proposition 1

*Proof.* Because the correlation between non-neighboring regions is sufficiently small and the correlation between neighboring regions is positive, we can simplify our proof by setting $\Sigma_{ii'} = 0$ for any two non-neighboring regions. Under such a condition, we have

$$\frac{N}{4}SC = \sum_{ii'}\Sigma_{ii'} - \sum_{j\neq k}\sum_{i\in\mathcal{C}_j}\sum_{i'\in\mathcal{C}_k}\Sigma_{ii'}$$

$$= \sum_{ii'}\Sigma_{ii'} - \sum_{j\neq k}\sum_{i\in\partial\mathcal{C}_j}\sum_{i'\in\partial\mathcal{C}_k}\Sigma_{ii'}W_{ii'},$$

$$\frac{N}{4}I_1 = 2\sum_{j\neq k}\sum_{i\in\partial\mathcal{C}_j}\sum_{i'\in\partial\mathcal{C}_k}\Sigma_{ii'}W_{ii'}.$$

Thus minimizing $SC + I_1$ is equivalent to minimizing

$$\sum_{j\neq k}\sum_{i\in\partial\mathcal{C}_j}\sum_{i'\in\partial\mathcal{C}_k}\Sigma_{ii'}W_{ii'}. \tag{4}$$

Under the global design, there has only one cluster, making the expression in (4) equal to zero. Given that all elements $\Sigma_{ii'}$ are non-negative, we conclude that the global design achieves the minimum value for (4).

For the second-order interference $I_2$, we analyze its three components separately to demonstrate that the global design is optimal:

- For $J_1$, when the $i, i'$-th regions are within the same cluster $\mathcal{C}_j$, we have $m_{ii'} \geq 1$, and thus $\frac{\Sigma_{ii'}(1-p^{m_{ii'}-1})}{p^{1+m_{ii'}}} \geq 0$. Therefore, $J_1$ is minimized under the global design where there are no boundaries.

- For $J_2$, we have

$$NJ_2 = \sum_{j\neq k}\sum_{i\in\partial\mathcal{C}_j}\sum_{i'\in\partial\mathcal{C}_k}\Big(\frac{\Sigma_{ii'}}{p^{m_{ii'}+1}} - \frac{2\Sigma_{ii'}}{p^2}\Big)\mathbb{I}(\mathcal{N}_{i'}\cap\mathcal{C}_j\neq\emptyset)\mathbb{I}(i, i' \text{ are two neighboring regions}).$$

Since $m_{ii'} \geq 2$ when $i, i'$ are adjacent, together with the fact that $p = 0.5$, $\frac{\Sigma_{ii'}}{p^{m_{ii'}+1}} - \frac{2\Sigma_{ii'}}{p^2}$ must be a non-negative value. The results ensures a global design minimizes $J_2$.

- For $J_3$, because $\mathcal{N}_{i'} \cap \mathcal{C}_j \neq \emptyset$ implies $m_{ii'} \geq 1$, then we have $\mathbb{I}(m_{ii'} > 0) \geq \mathbb{I}(\mathcal{N}_{i'} \cap \mathcal{C}_j \neq \emptyset)$, which ensures $\frac{\Sigma_{ii'}}{p^{m_{ii'}+1}} (\mathbb{I}(m_{ii'} > 0) - \mathbb{I}(\mathcal{N}_{i'} \cap \mathcal{C}_j \neq \emptyset))$ is always non-negative. This fact implies $J_3$ is minimized under the global design.

Since $\mathrm{SC} + I_1$ and $I_2$ are each minimized under the global design, their sum is also minimized under the global design. $\qquad \square$

### B.3. Proof of Proposition 2

*Proof.* When SUTVA holds, the interference terms $I_1$ and $I_2$ equal zero. For SC, we have

$$\mathrm{SC} = \frac{4}{N} \Big[ \sum_{ii'} \Sigma_{ii'} - \sum_{j \neq k} \sum_{i \in \mathcal{C}_j} \sum_{i' \in \mathcal{C}_k} \Sigma_{ii'} \Big]. \tag{5}$$

Thus minimizing MSE is equivalent to maximizing $\sum_{j \neq k} \sum_{i \in \mathcal{C}_j} \sum_{i' \in \mathcal{C}_k} \Sigma_{ii'}$, which equals the summation over correlations between different clusters. Due to the non-negativeness of $\Sigma_{ii'}$, it is maximized when $m = R$, where each region belongs to a unique cluster, i.e., as in the individual design. $\qquad \square$

### B.4. Proof of Proposition 3

*Proof.* As proved in Section B.1, we have

$$
\begin{aligned}
I_1 &= \frac{8}{N} \Big[ \sum_{i \in \mathcal{C}_1} \sum_{i' \in \partial \mathcal{C}_2} \Sigma_{ii'} \mathbb{I}(\mathcal{N}_{i'} \cap \mathcal{C}_1 \neq \emptyset) + \sum_{i \in \mathcal{C}_2} \sum_{i' \in \mathcal{C}_1} \Sigma_{ii'} \mathbb{I}(\mathcal{N}_{i'} \cap \mathcal{C}_2 \neq \emptyset) \Big] \\
&\leq \frac{8}{N} \Big[ \sum_{i \in \mathcal{C}_1} \sum_{i' \in \mathcal{C}_2} \Sigma_{ii'}^+ \sum_{\ell \in \mathcal{C}_1} W_{\ell i'} + \sum_{i \in \mathcal{C}_2} \sum_{i' \in \mathcal{C}_1} \Sigma_{ii'}^+ \sum_{\ell \in \mathcal{C}_2} W_{\ell i'} \Big] \\
&\leq \frac{8}{N} \Big[ \sum_{i \in \mathcal{C}_1} \sum_{i' \in \mathcal{C}_2} \sum_{\ell \in \mathcal{C}_1} \Sigma_{\ell i'}^+ W_{\ell i'} + \sum_{i \in \mathcal{C}_2} \sum_{i' \in \mathcal{C}_1} \sum_{\ell \in \mathcal{C}_2} \Sigma_{\ell i'}^+ W_{\ell i'} \Big] \\
&= \frac{8}{N} \Big[ |\mathcal{C}_1| \sum_{i' \in \mathcal{C}_2} \sum_{\ell \in \mathcal{C}_1} \Sigma_{\ell i'}^+ W_{\ell i'} + |\mathcal{C}_2| \sum_{i' \in \mathcal{C}_1} \sum_{\ell \in \mathcal{C}_2} \Sigma_{\ell i'}^+ W_{\ell i'} \Big], \tag{6}
\end{aligned}
$$

where the second inequality holds under Assumption 2. The last term (6) equals

$$\frac{8}{N} \Big( |\mathcal{C}_1| + |\mathcal{C}_2| \Big) \sum_{i' \in \mathcal{C}_1} \sum_{\ell \in \mathcal{C}_2} \Sigma_{\ell i'}^+ W_{\ell i'} = \frac{8R}{N} \sum_{i' \in \mathcal{C}_1} \sum_{\ell \in \mathcal{C}_2} \Sigma_{\ell i'}^+ W_{\ell i'},$$

which, together with Equation (5), gives us the objective function:

$$\frac{8}{N} \sum_{i \in \mathcal{C}_1} \sum_{i' \in \mathcal{C}_2} \Big[ R \Sigma_{ii'}^+ W_{ii'} - \Sigma_{ii'} \Big].$$

$\qquad \square$

**B.5. Proof of Proposition 4**

*Proof.* As stated in Section B.3, we have

$$
\begin{aligned}
\frac{8R}{N} \sum_{i' \in \mathcal{C}_1} \sum_{\ell \in \mathcal{C}_2} \Sigma_{\ell i'}^+ W_{\ell i'} &= \frac{8}{N} \Big[ \sum_{i \in \mathcal{C}_1} \sum_{i' \in \mathcal{C}_2} \sum_{\ell \in \mathcal{C}_1} \Sigma_{\ell i'}^+ W_{\ell i'} + \sum_{i \in \mathcal{C}_2} \sum_{i' \in \mathcal{C}_1} \sum_{\ell \in \mathcal{C}_2} \Sigma_{\ell i'}^+ W_{\ell i'} \Big] \\
&= \frac{8}{N} \Big[ \sum_{i \in \mathcal{C}_1} \sum_{i' \in \partial\mathcal{C}_2} \sum_{\ell \in \mathcal{C}_1} \Sigma_{\ell i'}^+ W_{\ell i'} + \sum_{i \in \mathcal{C}_2} \sum_{i' \in \partial\mathcal{C}_1} \sum_{\ell \in \mathcal{C}_2} \Sigma_{\ell i'}^+ W_{\ell i'} \Big] \\
&\le \frac{8}{N} \max_{i \ne i'} \Sigma_{ii'} \Big[ \sum_{i \in \mathcal{C}_1} \sum_{i' \in \partial\mathcal{C}_2} \sum_{\ell \in \mathcal{C}_1} W_{\ell i'} + \sum_{i \in \mathcal{C}_2} \sum_{i' \in \partial\mathcal{C}_1} \sum_{\ell \in \mathcal{C}_2} W_{\ell i'} \Big] \\
&\le \frac{8}{N} d_{\max} \max_{i \ne i'} \Sigma_{ii'} \Big[ \sum_{i \in \mathcal{C}_1} \sum_{i' \in \partial\mathcal{C}_2} \mathbb{I}(\mathcal{N}_{i'} \cap \mathcal{C}_1 \ne \emptyset) + \sum_{i \in \mathcal{C}_2} \sum_{i' \in \partial\mathcal{C}_1} \mathbb{I}(\mathcal{N}_{i'} \cap \mathcal{C}_2 \ne \emptyset) \Big] \\
&\le \frac{8}{N} d_{\max} \sigma \Big[ \sum_{i \in \mathcal{C}_1} \sum_{i' \in \partial\mathcal{C}_2} \Sigma_{ii'} \mathbb{I}(\mathcal{N}_{i'} \cap \mathcal{C}_1 \ne \emptyset) + \sum_{i \in \mathcal{C}_2} \sum_{i' \in \partial\mathcal{C}_1} \Sigma_{ii'} \mathbb{I}(\mathcal{N}_{i'} \cap \mathcal{C}_2 \ne \emptyset) \Big] \\
&= d_{\max} \sigma I_1,
\end{aligned}
$$

where the second equation holds because $W_{\ell i'} = 0$ when $i'$ falls in the interior region of its cluster, and the second inequality holds because each $i' \in \partial\mathcal{C}_k$ (for $k = 1, 2$) can have at most $d_{\max}$ adjacent neighbors that fall in $\mathcal{C}_j$. $\qquad\square$

## C. Experiments: details and additional results

### C.1. Synthetic environments

**Settings.** The model for the outcome is represented as follows:

$$
Y_{it} = 3O_{it} \sin\big(l_x + l_y + s \times (A_{it} + 0.5\bar{A}_{it})\big) + e_{it},
$$

where $O_{it} \in \mathbb{R}$ represents the covariates, $\bar{A}_{it} = \frac{1}{|\mathcal{N}_i|-1} \sum_{j \ne i, j \in \mathcal{N}_i} A_{jt}$ represents the average of neighboring treatments, and $s$ denotes the signal strength, which is set to 2.5% for our experiments. For each $t$, $(e_{1t}, \ldots, e_{Rt})$ are independently drawn from a zero-mean multivariate Gaussian distribution with time-invariant covariance $\Sigma$. Under the spatial setting, $\Sigma$ is set as one of the following correlation structures: (i) constant correlation, $\Sigma_{ij} = \rho^{\mathbb{I}(i \ne j)}$; (ii) truncated constant correlation, $\Sigma_{ij} = \mathbb{I}(i = j) + (\rho - R^{-1}|i - j|)\mathbb{I}(|i - j| \le \rho R)$; (iii) exponential correlation, $\Sigma_{ij} = \rho^{|i-j|}$. After structuring the regions, the corresponding synthetic environment is created to simulate the conditions that allow us to generate datasets.

We evaluate each method using the relative MSE as criterion. Specifically, we use the relative MSE which is computed as $\mathcal{R}^{-1} \sum_{r=1}^{\mathcal{R}} (\widehat{\mathrm{ATE}}_r - \mathrm{ATE})^2 / (\mathrm{ATE})^2$, where $\widehat{\mathrm{ATE}}_r$ is the estimated ATE returned by one estimator at the $r$-th independent replication. We set the total number of replications to $\mathcal{R} = 50$.

### C.2. Real-data simulator

Orders and drivers are simulated as follows:

1. New orders are generated according to historical data. These orders, along with existing unassigned orders, are processed and assigned according to the dispatch algorithm described in Tang et al. (2019).
2. Drivers who are assigned to orders will decide whether to accept them based on probabilities generated by a pre-trained LightGBM model. This step considers various characteristics of both the drivers and the orders.
3. Drivers who are idle and not currently assigned to any orders are directed to specific areas based on historical data of idle driver movements.
4. Drivers who are subject to repositioning must follow the directives provided by a pre-trained repositioning algorithm implemented by the ridesharing platform.
5. Once drivers accept orders, they proceed to the pickup locations, collect the passengers, and then travel to the specified destinations.
6. The pool of available drivers is continuously updated to reflect new drivers entering the service area and existing drivers who go offline.

Different from the synthetic environment, we cannot obtain the oracle ATE value by mathematical calculation. Therefore, we use a Monte Carlo (MC) procedure to approximate the true ATE value. Specifically, we run the simulator for $N$ independent days under actions 1 and 0, respectively, leading to the reward observations $\{r_{it}^{(1)}\}_{1 \le i \le R, 1 \le t \le N}$ and $\{r_{it}^{(0)}\}_{1 \le i \le R, 1 \le t \le N}$. With the two datasets, the MC estimator for ATE is given by:

$$\widehat{\text{ATE}}^{\text{MC}} = \frac{1}{N} \sum_{t=1}^{N} \sum_{i=1}^{R} \left[ r_{it}^{(1)} - r_{it}^{(0)} \right].$$

When $N \to \infty$, the $\widehat{\text{ATE}}^{\text{MC}}$ converges to ATE. In our experiment, we set $N$ to a large value and the obtained $\widehat{\text{ATE}}^{\text{MC}}$ is set as the surrogate of the true ATE. Specifically, $N$ is set as 1000.

### C.3. Implementation details

Algorithm 1 in the main text skips the estimation procedure of $\{g_i\}_{i=1}^{R}$ so as to simplify the illustration of the algorithm. Here, we elaborate the details. Besides, to circumvent the need for imposing stringent metric entropy conditions on the outcome regression function (Dai et al., 2020), we adopt the data-splitting and cross-fitting method (Chernozhukov et al., 2018).

The idea of cross-fitting is presented below. Given the repeated observations $\{(Y_{i,t}, A_{i,t}, O_{i,t})\}_{1 \le i \le R, 1 \le t \le N}$, we split the data into $K$ non-overlapped subsets of equal size. Then we estimate the regression function $g_i$ based on the training data $\{(Y_{i,t}, A_{i,t}, O_t) : t \notin \mathcal{I}_k, 1 \le i \le R\}$, where $\mathcal{I}_k$ denote the indices of the $k$th subset. We denote the estimated function $g_i$ as $\widehat{g}_i^{(k)}$ under such step. Under the cross-fitting framework, we denote the DR estimator as $\widehat{\text{ATE}}^{\text{DR-CF}}$. We summarized the complete algorithm for this in Algorithm 2.

---

**Algorithm 2** Estimating ATE at the $l$-th iteration using cross-fitting.

**Input:** Datasets $\mathcal{D}^{(j)} = \{(\boldsymbol{Y}_t, \boldsymbol{A}_t, \boldsymbol{O}_t)\}_{t=1}^{B}$ for $j = 1, \ldots, l$, as collected in Algorithm 1.
1: Split dataset $\mathcal{D}^{(1)} \cup \cdots \cup \mathcal{D}^{(l)}$ into $K$ non-overlapped subsets of equal size. Let $\mathcal{I}_k$ denote the set of the indices for the $k$th subset.
2: **for** $k = 1, \ldots, K$ **do**
3:     Compute the estimated outcome regression functions $\big\{\widehat{g}_i^{(k)}\big\}_{i=1}^{R}$ based on the data tuples $\{(Y_{i,t}, A_{i,t}, O_{i,t})\} \mid t \notin \mathcal{I}_k, i \in \{1, \ldots, R\}\}$ using random forest.
4:     Compute $\widehat{\nu}_{i,t}^{(k)}(\boldsymbol{a}) = \widehat{g}_i^{(k)}(A_{i,t}, O_{i,t}) + \frac{T_{i,t}(\boldsymbol{a})}{\mathbb{E}[T_{i,t}(\boldsymbol{a})]}[Y_{i,t} - \widehat{g}_i^{(k)}(A_{i,t}, O_{i,t})]$ for all $i \in \{1, \ldots, R\}$ and $t \in \mathcal{I}_k$.
5: **end for**
6: Compute the ATE estimator:

$$\widehat{\text{ATE}}_l^{\text{DR-CF}} = \frac{1}{lB} \sum_{k=1}^{K} \sum_{t \in \mathcal{I}_k} \sum_{i=1}^{R} [\widehat{\nu}_{i,t}^{(k)}(\boldsymbol{1}) - \widehat{\nu}_{i,t}^{(k)}(\boldsymbol{0})]$$

**Output:** $\widehat{\text{ATE}}_l^{\text{DR-CF}}$.

---

We give more details on the Step 3 and Step 5 in Algorithm 2.

- In Step 3, instead of using the observed data at each region to fit regression model, we concatenate the observed data in all region, and augment the data by including the longitude and latitude as two additional features. The advantage of this procedure is that it shares the information across spatial regions, making the estimated outcome regression model can still fit data well when the number of repeated observations $N$ is small. The advantage of this procedure is demonstrated by Figure 8. We can see that, when sample size is small, such procedures (denoted as OCGC, CGC) are much better than fitting $g_i$ individually with dataset $\{(Y_{it}, A_{it}, O_{it})\}_{t=1}^{N}$ (denoted as OCGC-Local, CGC-Local).

- In Step 6, we need to calculate the term $\mathbb{E}(T_{it}(\mathbf{a}))$ for estimation. Because the treatment are randomly assigned and

independent to covariates, $\mathbb{E}(T_{it}(\mathbf{a}))$ has a close form expression that is written as:

$$\mathbb{E}[T_{it}(\mathbf{a})] = (p)^a (1-p)^{1-a} \prod_{k \neq j} \left[ 1 - \prod_{i \in \mathcal{N}_l} \mathbb{I}\left(i \notin \mathcal{C}_k\right) (p)^a (1-p)^{1-a} \right] + \prod_{i \in \mathcal{N}_l} \mathbb{I}\left(i \notin \mathcal{C}_k\right).$$

Using this expression can simplify the computing on $\widehat{\text{ATE}}$.

- In the algorithm, we use $\mathcal{D}^{(1)}, \ldots, \mathcal{D}^{(l)}$ together to refit machine learning model, rather than using the dataset obtained in a single step (i.e., $\mathcal{D}^{(l)}$ generated in the $l$-th step). This approach sufficiently leverages the collected data as reported in Figure 9. We use the logarithmic scale on values of $y$-axis for a clearer presentation. From Figure 9, leveraging all data (denoted as CGC) generally decreases the MSE when compared to using only one dataset $\mathcal{D}^{(l)}$ (denoted as CGC-ST).

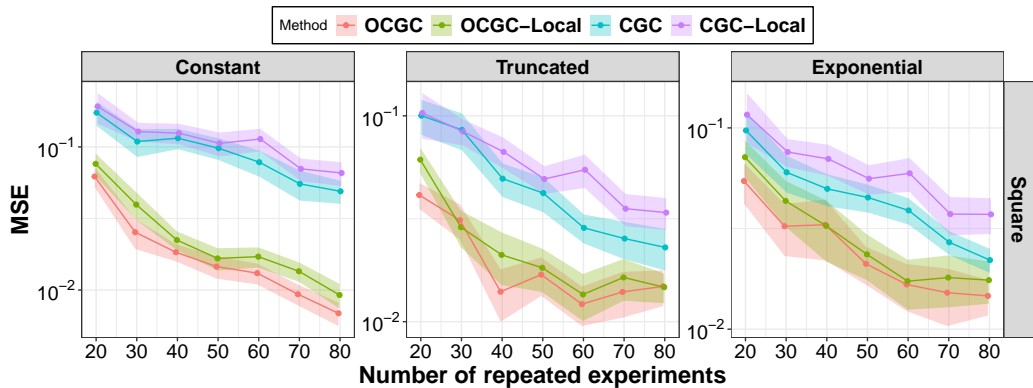

*Figure 8.* The plot of number of repeated experiments versus MSE on square spatial setting. Each panel corresponds to a spatial correlation setting. For clearer representation, we adopt a logarithmic scale for the values on the $y$-axis.

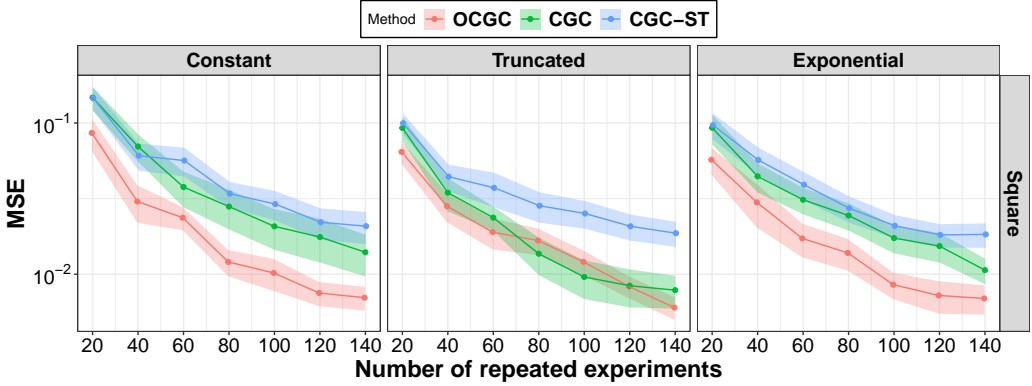

*Figure 9.* The plot of number of repeated experiments versus MSE on square spatial setting.

# D. Discussion and Future Works

Our method omits $I_2$ in the objective function to facilitate the optimization. We acknowledge that such second-order effects may be non-negligible. However, its inclusion would significantly increase the computational complexity. Developing a computationally tractable solution that properly accounts for this term remains a practical challenge for future work. In terms of applications, our methodology primarily targets settings where independent experiments can be repeatedly conducted over time. While this framework aligns well with our ridesharing application for spatial A/B testing (the focus of this paper), it may require adaptation for other settings. From the empirically effective performance of our approach on the single experiment, it would be a promising direction to extend our approach to more general experimental settings.

