# OpenReview forum: "Balancing Interference and Correlation in Spatial Experimental Designs: A Causal Graph Cut Approach"
_ICML.cc/2025/Conference — ICML 2025 poster_

### Official Review · Reviewer_Lhaw · 2025-03-12

**Overall Recommendation:** 2

**Summary:**

This paper studies optimal cluster-randomized designs for spatial A/B testing problems. The authors investigate the decomposition of the mean squared error (MSE) and show that interference and correlation are two key driving factors, contributing in opposite directions: when interference is strong, it is optimal to assign the same policies to neighboring regions, whereas strong correlation favors assigning different policies. They also propose a computationally efficient surrogate function for the MSE, which adapts to varying levels of interference and correlation structures. Combined with a graph cut algorithm, the surrogate function effectively learns the optimal design, as demonstrated by both theoretical analysis and experimental results.

**Claims And Evidence:**

The claims regarding the contributions in spatial A/B testing problems make sense to me, however, I believe the authors should provide detailed discussions if they would like to claim their method also *is readily adaptable to Example 2: Environmental and epidemiological applications* and *Example 3: Experimentation in social networks.*

**Essential References Not Discussed:**

I don't see any obvious missing references.

**Experimental Designs Or Analyses:**

Some of the simulation results make me worry about their credibility. Below, I list the concerns that I hope the authors can address in their response:
- In Figure 5(b), the MSE of the ID method also improves as the number of repetitions increases. Intuitively, however, it should remain the same, since it is always an individual design. Could the authors clarify this behavior?
- The theoretical results rely heavily on assumptions about the covariance matrix $\Sigma$. For example, the surrogate function is valid only when all entries are non-negative, and Assumption 2 requires a decaying covariance structure. I would like to know how the authors handle cases where the estimated covariance $\hat{\Sigma}$ does not satisfy these conditions. How does the algorithm deal with such situations, and what is the expected performance of the proposed method in these cases?

**Methods And Evaluation Criteria:**

They make sense to me, but I would appreciate it if the authors could conduct more experiments on real-world datasets.

**Other Comments Or Suggestions:**

In line 172, the right square bracket is misplaced.

**Other Strengths And Weaknesses:**

- Strengths: The paper is generally well written and easy to follow. The trade-off between interference and correlation is clearly articulated within the studied setting. Additionally, the intuition behind the design of the surrogate function is reasonable and well motivated.

- Weaknesses: My main concern lies in the assumptions made by the paper. Specifically, Assumption 2 and the conditions in Propositions 1 and 2 regarding the covariance matrix $\Sigma$ may limit the applicability of the proposed method. In theory, when these assumptions are violated, the trade-off between interference and correlation, as well as the surrogacy result in Proposition 3, no longer hold. In practice, it is unclear how the proposed method performs when the estimated covariance matrix $\hat{\Sigma}$ does not satisfy these conditions. Clarification or empirical evaluation in such scenarios would strengthen the paper.

**Questions For Authors:**

Please see my comments above in *Experimental Designs Or Analyses* and *Other Strengths And Weaknesses*.

**Relation To Broader Scientific Literature:**

This paper is related to causal inference and experimental design with spatial interference.

**Theoretical Claims:**

I reviewed the proof of Theorem 1, and it appears sound to me.

---

> ### Author Rebuttal · Authors · 2025-03-31
>
> >Applications to other data examples
>
> We sincerely appreciate your valuable feedback. In response, we conducted extensive investigations during the rebuttal, including: (1) empirical validation on additional data, (2) methodological extensions, (3) theoretical investigations. The methodological  investigation is elaborated in our response to Reviewer VLbx.
>
> **Empirical validation on environmental data**. During rebuttal, we found a paper that studies the causal effect of wind speed on PM10 levels (Zhang et al., 2024, arXiv:2409.04836, Section 6). As the raw dataset cannot be directly used for evaluation (refer to our first response to Reviewer 96AW), we utilized the simulator described in their paper for comparison. The results, shown in [Figure](https://www.dropbox.com/scl/fi/usk0pcpv7h615634c8b4k/Lhaw_pm10.pdf?rlkey=qkabneg3idllwi4xc4zje89e9&st=aei7qaq2&dl=0), demonstrate our proposal achieves smaller MSEs compared with the benchmarks, confirming its applicability to environmental applications.
>
> **Methodological extensions**. Our primary focus is Example 1, but we have extended our design to handle two additional settings: (i) a single-experiment setting without repeated measurements and (ii) a multi-experiment setting allowing the carryover effects over time. For the single-experiment setting, our approach remains effective given prior knowledge of the covariance structure, as confirmed by our newly conducted numerical study. For multi-experiment settings with carryover effects, we have outlined the methodology while reserving its implementation for future work, as detailed in our response to Reviewer VLbx. Collectively, these extensions enable our framework to accommodate the applications described in Examples 2 and 3.
>
> **Theoretical investigation**. We have identified crucial assumptions that guarantee the validity of our proposed extension in the multi-experiment setting (ii). Specifically, we would require a Markov assumption (Puterman, 2014, John Wiley & Sons) and a temporal mixing condition (Bradley, 2005, Probab. Surv.) to maintain the covariance estimator's consistency in the presence of temporal dependencies. Similar conditions are widely employed in the RL literature for consistent estimation (Kallus and Uehara, arXiv:1909.05850).
>
> >Covariance assumption
>
> We have comprehensively addressed your concern in the following three ways: First, we conducted extensive empirical validation under violations of the covariance assumption. Second, we clarified the role of the non-negativity assumption. Third, we developed methodological extensions for scenarios where Assumption 2 fails to hold.
>
> **Numerical experiments**: We conducted **another new experiment** during rebuttal to illustrate the robustness of our proposal. We design a periodic covariance function $\Sigma_{ij} = 1 - \rho \times ((i - j)\mod 3)$. Under this choice, the decaying covariance assumption no longer holds and the non-negativity assumption is violated when $\rho>0.5$. The results, shown [here](https://www.dropbox.com/scl/fi/n08067ujhj9if8fru1gq1/Lhaw_PeriodCov.pdf?rlkey=z6mo09pntxihowq295b64cuax&st=o8jrb7g2&dl=0), demonstrate that our methods still outperform baselines.
>
> We also remark that in both our **ridesharing simulator** and the **PM10 dataset** (see our first response), we find violations of the two assumptions. Nonetheless, our designs remain highly competitive. These results consistently demonstrate our proposal's robustness.
>
> **Non-negativity**: The non-negative assumption is primarily employed in Prop. 1 & 2 to demonstrate the trade-off between interference and correlation. It is not required to ensure the proposed surrogate function forms a valid upper bound (Prop. 3).
>
> **Methodological extension**: While the decaying covariance assumption (Assumption 2) is common in spatial statistics (Cressie, 2015), our method remains flexible when this condition is violated. In such cases, we propose a simple fix: scaling the first term in loss function (2) by a constant $C > 1$ to ensure it remains an upper bound. The hyperparameter can be optimally determined through simulation studies based on experimental data evaluating the performance of the resulting estimator across different $C$ values.
>
> >MSE of ID in Figure 5(b)
>
> Your are right that the **asymptotic** MSE of ID shall remain constant regardless of the number of repetitions $N$. However, in **finite sample**, the MSE actually improves with $N$, due to the estimation of the $g$ function.
>
> Specifically, in theory, Neyman orthogonality guarantees that the asymptotic MSE will match that with an oracle g given sufficiently large $N$. In practice, we estimate $g$ by incorporating all prior data at each experiment. This leads to an initial period where the MSE decreases as $N$ increases, reflecting the improvements in the estimation of $g$. As the estimated $g$ approaches its oracle value, the MSE then stabilizes. The trend shown in Figure 5(b) clearly demonstrates this pattern.

---

### Official Review · Reviewer_96AW · 2025-03-14

**Overall Recommendation:** 4

**Summary:**

This paper proposes a graph cut approach for design of spatial experiments to estimate Average Treatment Effect (ATE) in settings with both interference where SUTVA assumption does not hold as well as when there exists correlation between units. The authors present a method which builds flexible surrogate function for the Mean Squared Error (MSE) of the ATE estimator. Both theoretical empirical results corresponding to the proposed approach is presented.

**Claims And Evidence:**

To my understanding, the claims made in the paper are well supported with both theory and methodology. Experiments on real data simulators are welcome.

**Essential References Not Discussed:**

I think the paper does a good job in discussing the related literature.

**Experimental Designs Or Analyses:**

The experimental design and analyses on both synthetic data and simulator seem quite interesting and sound. The paper would have probably benefited if there was real data that could have been tested on.
One of the clarifications I would appreciate is for synthetic data how the outcome equation was chosen (line 906). Is it standard in prior works or is it based on some heuristic?

**Methods And Evaluation Criteria:**

The proposed method is interesting and certainly relevant for the problem at hand. Evaluation criteria seems fair with multiple different baselines included.

**Other Comments Or Suggestions:**

In figure 5 (b) it should be number of "repetitions".

**Other Strengths And Weaknesses:**

I think the paper does a good job of exposition of the idea and also the setting. However, I think it would be great if some of the limitations of the present approach was discussed in the paper. For example, the algorithmic complexity when the interference/ correlation is high, possible limitations of design procedure itself in practical settings and so on.

**Questions For Authors:**

See above.

**Relation To Broader Scientific Literature:**

Although I am not too familiar with broader scientific literature in this field, this paper proposes a method which can account for both interference and correlation between units in policy evaluation setting which I believe is not widely addressed in literature before. Although the proposed approach is simple, it addresses a problem which is practically relevant. Some of the prior works require SUTVA assumption.

**Theoretical Claims:**

I did not check in detail the correctness of proofs.

---

> ### Author Rebuttal · Authors · 2025-03-30
>
> We sincerely appreciate your thoughtful review and valuable feedback on our work. Your positive assessment is particularly encouraging to us. Below, we provide detailed responses to each of your comments.
>
> > **Real data analyses**
>
> While we do have a real-world dataset, it cannot be directly used for evaluation. To properly assess a given design, one would need to: (1) implement the design to generate corresponding data, and (2) compute the MSE from this generated data. This is precisely why we developed our real-data-based simulator, which enables adaptive data generation tailored to different experimental designs for  comparison.
>
> During the rebuttal, we identified another publicly available real-world dataset that examines the causal effect wind speed on PM10 levels [1]. However, as with our ridesharing dataset, this raw observational data cannot be directly used for evaluation. We therefore employed the simulation model from [1] to generate synthetic data under different designs. Our simulation results, visualized in [Figure](https://www.dropbox.com/scl/fi/usk0pcpv7h615634c8b4k/Lhaw_pm10.pdf?rlkey=qkabneg3idllwi4xc4zje89e9&st=wt8kuzxu&dl=0), demonstrate that our method achieves significantly lower MSEs compared to existing designs.
>
> > **Rationality of the synthetic outcome model (line 906)**
>
> Our outcome regression function follows the nonparametric model from [2] (Page 24), which studied the spatial A/B testing problem as well. We selected this model for two reasons: First, the outcome is a complex nonlinear function of observations, treatments, and spillover effects. Second, it naturally incorporates spatial heterogeneity through latitude/longitude coordinates, allowing for geographic variation. We consider these  features to approximate the complex dynamics present in real-world settings.
>
> > **Limitations**
>
> There are a couple of limitations of our proposal:
>
> * Theoretically, our characterization of the interference-correlation trade-off relies on the assumption of non-negative covariance functions. While in practice we expect the presence of some negative covariances would not alter our conclusions, it remains unclear how to elegantly relax this mathematical constraint while preserving our theoretical findings.
>
> * Methodologically, we omit the second-order interference term in the objective function to facilitate the optimization. We acknowledge that such higher-order effects may be non-negligible. However, its inclusion would significantly increase the optimization complexity. Developing a computationally tractable solution that properly accounts for this term remains a practical challenge for future work.
>
> * In terms of applications, our methodology primarily targets experimental settings where independent experiments can be repeatedly conducted over time. While this framework aligns well with our ridesharing application for spatial A/B testing (the focus of this paper), it may require adaptation for other settings. We have discussed modifications for scenarios with either a single experiment or multiple experiments with carryover effects in our response to Reviewer VLbx. Extending our approach to more general experimental settings would be a valuable direction for future research.
>
> > **Typos**
>
> Thanks for pointing them out. They will be corrected.
>
> [1] Zhang, W., et al. Spatial Interference Detection in Treatment Effect Model.
>
> [2] Yang, Y., et al. Spatially Randomized Designs Can Enhance Policy Evaluation.

---

### Official Review · Reviewer_NcNs · 2025-03-14

**Overall Recommendation:** 4

**Summary:**

The paper proposes a new method for experimental design in settings with repeated experiments, spatial interference and error correlation. The authors characterize the MSE of the doubly-robust average treatment effect estimator, and propose an upper bound objective function that depends on estimable quantities and can be minimized using standard graph cutting algorithms. Their proposed method uses information from the estimated error correlation structure to balance the contributions of cross-cluster and within-cluster correlation in the MSE, and can be computed efficiently. Through a synthetic simulation exercise and a real data simulation study the paper shows that the proposed method performs well relative to other commonly used methods in the literature.

**Claims And Evidence:**

The paper makes three claims relating their method to the literature. I find the claims that their method is more adaptable to correlation structures and computationally efficient reasonable and an interesting contribution. I find the claim that the paper offers a method that is more flexible and better suited for moderate/large interference effects than the literature (Viviano et al. 2023) more nuanced.

1. The paper would benefit by being more clear in how their assumptions relate to Viviano et al. 2023 and in particular how their objective differs from Viviano et al. 2023. In their paper both bias and variance are considered and some of their results and assumptions are aimed at minimizing bias, which may differ from the MSE goal of this paper.

2.  The paper considers a setting in which repeated experiments are available. The paper should be more clear about what is gained by having repeated experiments and distinguish the improvements from having repeated experiments versus not having them and their method versus the literature.

**Essential References Not Discussed:**

The paper considers the key references in the literature.

**Experimental Designs Or Analyses:**

The simulation design is sound. However, I have the following comments:

1. Figure 5 and Figure 6 suggest that the main improvement is to use repeated measurements. More discussion on whether the other methods used for comparison benefit from repeated measurements or not and why the differences are so stark would be helpful in understanding the benefits of the proposed method, specially given that ID and GD perform so much better than the other methods too.

2. It might be helpful to have a simulation design for which we expect the other methods proposed in the literature to work to compare with the proposed method.

3. Might be helpful to split MSE into the SC, $I_1$ and $I_2$ to see the contribution of each term.

**Methods And Evaluation Criteria:**

The proposed method is sensible and the evaluation criteria (MSE) and simulations exercises are well suited to study this topic. However, usually in the literature there is a focus on the bias-variance trade off, and it is well known that

"The choice of clustering must balance two competing objectives: the larger the clusters (and the smaller
the number of clusters), the smaller the bias of the estimated global effect, but the larger its
variance."  (Viviano et al. 2023)

While focusing on the MSE is intuitive, usually the worry with interference is that it will bias our estimator of interest and so commenting on the bias of different designs would be helpful. If the estimator is unbiased regardless of the interference pattern or if the bias is the same regardless of the assignment mechanism it would be useful to clarify this.

**Other Comments Or Suggestions:**

The paper has a few typos:

1. Bracket in 172.
2. Independently in 212
3. Clarifying that the boundary of the set $\mathcal{C}$ is with other clusters.
4. In Assumption 2 "such that only"
5. Capital W in 373
6. Plots in Figure 5 and 6 show same estimators in different colors which is confusing

**Other Strengths And Weaknesses:**

I enjoyed reading the paper, it is well written, clear and provides a new interesting method.

**Questions For Authors:**

Beyond the comments above:

1. The definition of ATE is a bit odd in that it is not the average over $R$ but the sum. Might be good to comment on this as other papers, like Viviano et al. 2023, consider the average and do the asymptotics with respect to R.
2. What are the assumptions on $O_i$. Should we expect that it is independent across experiments? Same question with $e_i$.

**Relation To Broader Scientific Literature:**

Relative to the broader literature, and in particular Viviano et al. 2023, the paper provides a computationally efficient method that uses the error correlation structure to improve the MSE performance of ATE estimators by utilizing repeated experiments. This is new and interesting contribution to the literature when repeated experiments are available, but its specific merits should be further clarified.

**Theoretical Claims:**

The theoretical results are well stated and the proofs appear mostly correct. However, I have a couple questions/comments for which I need further clarification.

1. The $I_1$ term of Theorem 1 depends on the correlation between units at the boundary of one cluster with all other units in the other cluster. I was surprised to see that this can be upper bounded by a term that depends only on the boundary regions of each cluster (as $W_{ii}$ ensures in formula (2)).

2. It would be helpful to state if there are any restrictions between O, g, and the error term e. Under which conditions should we expect algorithm 1 to consistently estimate all the elements in the covariance matrix? Are we ruling out that the nature of the interference is similar to the error correlation structure?

3. The paper claims that the results dont require the fraction of boundary points to go to zero, but then ignores $I_2$ wouldnt this be a similar assumption to Viviano et al?

---

> ### Author Rebuttal · Authors · 2025-03-31
>
> We greatly appreciate your constructive comments and your positive assessment of our paper. We focus on your major comments below.
>
> >Comparison with Viviano et al. (2023)
>
> **Difference in objective**: One of the key difference lies in the choice of the estimator. Specifically, our estimator explicitly accounts for interference and remains unbiased regardless of the interference pattern (we will make this clearer in the paper), whereas Viviano et al.'s IS estimator is biased under interference. As such, although both minimize MSE, our optimization focuses exclusively on variance reduction whereas they must jointly consider both bias and variance.
>
> **Gain from repeated experiments**: Having repeated experiments allows us to accurately estimate the covariance function, which is crucial for our approach to achieve adaptivity. In comparison, such estimation is not feasible in Viviano et al.'s setting. As such, they adopted a minimax formulation that considers the worse case across all covariance functions.
>
> Following your suggestion, we conducted an ablation study during the rebuttal to distinguish the improvements from having repeated experiments. We consider single-experiment settings with certain prior knowledge regarding the covariance function, e.g., having access to a noisy covariance matrix. We kindly refer you to our response to Reviewer VLbx for the experimental results (see **Extensions to Single-Experiment Settings**).
>
> >Theoretical claim 1
>
> The inequality holds due to the presence of $R$ in (2) and the indicator function $\mathbb{I}(\mathcal{N}\_{i'} \cap \mathcal{C} \neq \emptyset)$ in $I_1$. The key step in establishing this inequality lies in upper bounding the indicator by $\sum_{\ell \in \mathcal{C}} W_{\ell i'}$. This inequality holds because when $i'$ is adjacent to $C$, there exists at least one $i\in \mathcal{C}$ such that $W_{ii'}=1$; see this [Figure](https://www.dropbox.com/scl/fi/r4zqqzmruzge7bu5zyklz/Ncns_Theory.pdf?rlkey=3talrjlyq4v2zdxnozv9dzsim&st=7j7r4x4y&dl=0) for a graphical illustration. When restricting to two clusters, this leads to bounding $I_1$ by two triple sums $\sum_{i\in \mathcal{C}\_1}\sum_{i'\in \mathcal{C}\_2}\Sigma_{ii'}^+\sum_{\ell'\in \mathcal{C_1}}W_{\ell i'}$ and $\sum_{i\in \mathcal{C}\_2}\sum_{i'\in \mathcal{C}\_1}\Sigma_{ii'}^+\sum_{\ell'\in \mathcal{C_2}}W_{\ell i'}$. The outermost summation in each triple (over $i$) produces terms proportional to the cluster sizes $|\mathcal{C}\_1|$ and $|\mathcal{C}\_2|$, whose sum adds up to $R$ (the number of regions). As such, the sum of all units in the other clusters is accounted by the factor $R$ rather than terms that depend only on the boundary regions.
>
> >Theoretical claim 2 & Question 2
>
> Three conditions are needed to consistently estimate covariance:
>
> * Each $g_i$ can be consistently estimated;
> * Each error $e_i$ is additive, i.e., independent of the covariates and treatment;
> * The error-covariates pairs are independent across experiments.
>
> *Note*: The independence condition (3) can be relaxed to certain mixing condition that allows for temporal dependence, provided the dependence decays sufficiently quickly over time (Bradley, 2005, Probability Surveys).
>
> >Theoretical Claims 3
>
> There are some differences between the two approaches, which we clarify below. In our proposal, we remove only the high-order interference term, and keep the first-order interference term in the objective function. To the contrary, Viviano et al imposes a weak interference assumption that removes all relevant interference terms when calculating the worst-case variance (see their proof of Lemma 3.2).
>
> >New experiments
>
> The baselines' inferior performance partially stems from their reliance on IS, while our method, ID and GD employ DR. To mitigate this effect, we conducted a new experiment by increasing variance of the random error, so as to reduce the impact of variance reduction by DR. [Results](https://www.dropbox.com/scl/fi/8rtxjnuj58an93z5pbc94/Ncns_Experiment2.pdf?rlkey=mueh74ksm8lxlao76w6ywwffu&st=lxfsa0yy&dl=0) show that the modification reduces (but does not close) the performance gap between our method and the baselines compared to Figs 5 & 6.
>
> Meanwhile, we conducted another experiment under a combination of (i) various covariance structures, (ii) the magnitude of correlation $\rho$, and (iii) the number of grids $R$ to report the contributions of SC, $I_1$, $I_2$. [Results](https://www.dropbox.com/scl/fi/x1wk3ck2x9goq2rphcrwu/Ncns_Experiment3.pdf?rlkey=rvgdg0n86i3r4ox5u1essfiii&st=53swq9tn&dl=0) suggest that $I_2$ can be proportional to $I_1$. This is because the plots visualize the MSE of our estimated optimal design rather than the oracle optimal design. Since our objective function involves only $I_1$ and not $I_2$, it reduces $I_1$ by increasing $I_2$, making $I_2$ comparable to $I_1$.
>
> >Question 1
>
> Will revise the definition as an average over $R$ regions to maintain consistency with the literature.

---

> > ### Comment · Reviewer_NcNs · 2025-04-03
> >
> > Thank you for your careful responses and for the additional experiments. I am still slightly confused as to why ID and GD perform so much better than the other methods also in the case with a single experiment that your provided during the rebuttal. I think clarifying why your method works better than the alternatives in the single experiment setting but the others do not would help clarify the properties of your method relative to the literature.

---

> > > ### Author Response · Authors · 2025-04-06
> > >
> > > Thank you for bringing up your question. We appreciate the opportunity to further clarify your confusion.
> > >
> > > **Comparison between ID, GD and the baseline designs**. In the single-experiment scenario, ID and GD outperform the three baseline estimators because these baseline methods rely on IS for ATE estimation, while ID and GD utilize DR (as in our proposal) to reduce the variance of IS. To illustrate this, we conducted additional simulation studies to replace the DR estimators in GD and ID with IS estimators. As shown in the [results](https://www.dropbox.com/scl/fi/y93rkdvj0akwl4pctnyo2/NcNs_IS.pdf?rlkey=slh37ys3qet4fdgeggepm56lm&st=2ewx585e&dl=0), both GD and ID perform noticeably worse than the three baseline estimators.
> > >
> > > **Comparison between our design and the baseline designs**. There are two key advantages of our design over the baseline designs:
> > >
> > > * Unlike the baseline designs, which are derived from IS, our design is directly derived from DR. DR is expected to perform better than IS in terms of ATE estimation.
> > > * Our design is adaptive to different spatial covariance functions, while the baseline designs do not adapt and instead typically rely on a minimax formulation. In multi-experiment settings, our method achieves adaptivity by estimating the covariance function using data from previous experiments. In single-experiment settings, however, adaptivity requires certain prior knowledge about the covariance function.
> > >
> > > During the rebuttal, we conducted additional simulations that increased the variance of the error term to reduce the variance-reduction effect of DR and to demonstrate the second advantage (see our response in the New experiments section). To further demonstrate this advantage, during this round, we also applied DR for ATE estimation in the three baseline designs, despite them being originally derived from IS. The results show that the second advantage is particularly valuable in settings with non-stationary covariance functions. Specifically, it can be seen from [Figure](https://www.dropbox.com/scl/fi/gm1wpfo5p9qtqcwrf7wj6/Ncns_DR.pdf?rlkey=t23hksuzv5yakqe619lj7dpb5&st=hck9gl5a&dl=0) that our method produces superior clusters than the three baseline methods, consistently resulting in lower MSEs. Meanwhile, ID performs worse than the three baseline methods. As for GD, it was not included in this comparison because it cannot identify the function $g$ in the single-experiment setting — where only treatment or control data is available, but not both. In our previous response, GD was included by assuming $g$ to be a constant function of the treatment, which yields a relatively small MSE in cases with weak average treatment effects. It does not work when either treatment effects are large, or IS is used for ATE estimation (as evidenced in our first comparison).

---

### Official Review · Reviewer_VLbx · 2025-03-20

**Overall Recommendation:** 2

**Summary:**

This work addresses the problem of experimental design under interference. The authors assume that the interference structure can be accounted for by a covariance matrix which is proposed to be estimated from data via repeated experimentation. The key insight of this work is a decomposition of the global average treatment effect in terms which decomposes intrinsic variance and direct and indirect network components. This decomposition motivates the development of a graph cut based algorithm, with bernoulli randomization performed over clusters. Empirical evidence provided via synthetic data an simulations show strong performance in comparison with other commonly used methods.

**Claims And Evidence:**

Yes, overall I think the claims are well laid out, the evidence is also good, though I would have liked to have seen a slightly larger set of empirical results.

**Essential References Not Discussed:**

Should probably cite the work on experimental design (see above), as well as Fatemi, Zahra, and Elena Zheleva. "Minimizing interference and selection bias in network experiment design." Proceedings of the International AAAI Conference on Web and Social Media. Vol. 14. 2020.

**Experimental Designs Or Analyses:**

Yes, I did. As I mention above, I would have liked to see a larger range (in particular varying topologies, etc. ) but what was performed is sound.

**Methods And Evaluation Criteria:**

The authors rely on sample splitting/cross fitting in order to estimate the outcome model. However, I don't see how we can assume we are able to effectively perform cross fitting without a lot more additional considerations. The authors assume that we are able to conduct the same experiment multiple times. This is a very strong assumption on multiple levels. For many settings in both industry and the social sciences it is infeasible to run the same experiment multiple times. Even when we are able to run the experiment multiple times we are assuming that we can assume independence across time, i.e., that having received treatment $i$ at round $t$ has no impact on future outcomes (which often fails to hold). If instead of assuming that we are running an experiment multiple times on the same population we are assuming that we are subsampling a population and running an experiment, we are in the regime of sampling independent but representative samples from a network, a highly nontrivial problem which will induce many of the problems the paper is seeking to avoid. Given this, it's not clear how this method works outside of bespoke settings without being able to have oracle access to the network structure. I'm curious if I've missed something or if the authors have a set of scenarios where we'd expect to have this structure.

**Other Comments Or Suggestions:**

Please see above regarding the repeated experiments.

**Other Strengths And Weaknesses:**

I think the decomposition is nice. The insight has existed implicitly in the literature but it is useful to see it explicitly spelled out.

**Questions For Authors:**

See above, also in design it is known (see Kallus) that without further assumptions on the potential outcomes Bernoulli randomization is minimax optimal. It would be good to have this mentioned in the paper. My question is how we should think about how that assumption interplays with the necessary assumptions for estimating the GATE.

**Relation To Broader Scientific Literature:**

This paper builds on a growing literature on network experimentation. The key insight sits between work in the experimental design literature (e.g., "Optimal A Priori Balance in the Design of Controlled Experiments" by Kallus, the Gram-Schidt Walk paper by Harshaw et al.) which seek to find designs where treatment status is negatively correlated with feature distance, and network experiementation (e.g., Ugander et al), where clustering is required to make plausible inference on global treatment effects.

**Theoretical Claims:**

Yes. I found the claims to be well founded.

---

> ### Author Rebuttal · Authors · 2025-03-31
>
> > Methods And Evaluation Criteria
>
> We appreciate your thoughtful comments, which are mainly about our settings with repeated and independent experiments. We believe your concerns may arise from certain misunderstandings of our paper, and we appreciate the opportunity to clarify them. We address them in three ways: (1) clarifying our focus on spatial A/B testing; (2) demonstrating that our approach is readily applicable to single-experiment settings, supported by promising numerical results; (iii) showing that our approach can be extended to handle carryover effects (the delayed treatment effect you mentioned).
>
> (1) **Focus of this paper**. We clarify that this paper focuses on the **spatial** setting -- specifically, Example 1 (A/B testing in marketplaces) -- and **not network** A/B testing. While many designs from network experimentation can be adapted to our setting (and we compared them numerically), our primary focus, as stated on Page 1 (the last sentence), remains Example 1.
>
> In applications like **ridesharing**, experiments can be conducted daily, with data across days treated as independent. This is due to the drop in demand early in the morning (1 -- 5 AM), ensuring each day’s data represents an independent realization. Such settings are well-adopted in prior work (Li et al., 2023, NeurIPS; Li et al., 2024, ICML; Luo et al., 2024, JRSSB). Another application occurs in **marketing auctions**, where daily budget resets eliminate carryover effects, making the independence assumption plausible (Basse, 2016, AISTATS; Liu, 2020, arXiv:2012.08724).
>
> (2) **Extensions to single-experiment settings**. Our core methodology does not rely on this assumption of repeated experiments. It remains applicable to single-experiment settings when we have certain prior knowledge regarding the underlying covariance matrix, either from a pilot study or based on historical data.
>
> To satisfy the practical requirement, we utilized a proxy covariance matrix, obtained by inserting noises into the true covariance matrix, and conducted additional experiments during the rebuttal. The [results](https://www.dropbox.com/scl/fi/cnbq91x32ygqfj7k6kb1t/VLbx_NoRepeat.pdf?rlkey=8rn9ual6o8nv2qbbhzrtx2ds9&st=cg00de5r&dl=0) demonstrate that our estimator: (i) maintains optimality against existing methods, (ii) achieves near-oracle performance (comparable to the oracle method with the true covariance matrix), (iii) remains robust to the approximation errors.
>
> We also remark that while cross-fitting helps simplify the theory (by avoiding the need for imposing VC-class conditions on $g$ to establish the asymptotics of the ATE estimator), our method remains effective without it - numerical results above were obtained without cross-fitting.
>
> (3) **Extensions with carryover effects**. The target here becomes the cumulative ATE aggregated over time. To handle carryover effects, we can use existing doubly robust estimators from the RL literature (Kallus & Uehara, 2022, OR); see also first equation on Page 18 of Yang et al. (2024). The resulting estimator maintains a similar form to ours, with two modifications: (i) the function g is replaced by a Q-function, and (ii) the IS ratio is replaced by its marginalized counterpart (Liu et al., 2018, NeurIPS). Treatments over time can be assigned using switchback designs, which are widely adopted (Bojinov, 2023, MS; Xiong et al., 2024, arXiv:2406.06768). As shown in Theorem 5 of Yang et al. (2024), the MSE of this ATE estimator has a similar closed-form expression, enabling us to apply the same decomposition and design a similar surrogate loss for optimization.
>
> > Experimental Designs or Analyses
>
> In the rebuttal, we considered other topologies and conducted additional experiments. Results reported [here](https://www.dropbox.com/scl/fi/hrhn52rrbavnrt9heokaz/Vlbx_Topologies.pdf?rlkey=3zcr40ukuyysaj3z8x3flycdo&st=2h15akph&dl=0) demonstrate the advantages of our proposal over baselines.
>
> > Essential References
>
> We are happy to include these additional references and discuss their findings as you required. However, as we discussed, we primarily focus on the **spatial** setting, and references on network experimentation might not be essential.
>
> > Bernoulli randomization
>
> The Bernoulli randomization is the same as our individual design. While Kallus (2018) established its minimax optimality without additional assumptions, we also proved its optimality in Proposition 2 without spatial interference. However, in settings with spatial interference, such a design is no longer guaranteed to be optimal. Specifically, Theorem 3.4 in Viviano et al. 2023 shows that cluster randomization outperforms Bernoulli randomization when the interference is not too weak. We also provide theoretical (Proposition 1) and numerical (extensive experiments) evidence that Bernoulli randomization becomes suboptimal in such cases.

---

### Decision · Program_Chairs · 2025-05-01

**Decision:**

Accept (poster)

**Comment:**

The authors consider the spatial interference problem for causal inference. The key observation is a graph-cut based experimental design that uses MSE decompositon in terms of intrinsic variance and spatial effects. They propose an experimental design algorithm based on these observations and evaluate their method via simulations and a real world simulator.

One expert reviewer is concerned about the assumption that the authors are performing the same experiment multiple times, calling it a very strong assumption on multiple levels. Unfortunately, the reviewer did not explicitly comment on the author's rebuttal nor participated in the AC discussions. Based on my own reading of the authors' rebuttal to the reviewers' comments, I can say that they did a good job. The authors emphasize the difference between spatial and network interference settings. They also suggest that their method can handle the single-experiment setting, so long as some additional assumptions are satisfied. Finally, they provide additional experimental results to supplement these claims.

The other critical reviewer was concerned about the assumptions that led to the theory. The authors, in their rebuttal, conducted additional experiments testing the robustness of their method to relaxations of this assumption. The reviewer did not respond to the authors nor participated in the AC discussions but I find the authors' reply satisfying for the questions raised by this reviewer.

The positive reviewers appreciated the contribution, such as computational efficiency relative to related work, and clarity of writing. Even the reviewers with weak reject scores acknowledged the contribution of the paper.